# JsonTuning: Towards Generalizable, Robust, and Controllable Instruction Tuning

## Abstract

Instruction tuning has emerged as a crucial process for harnessing the capabilities of large language models (LLMs) by providing explicit task instructions, leading to improved performance in various tasks. However, prevalent text-to-text instruction tuning (TextTuning) methods suffer from limitations in generalization, robustness, and controllability due to the ambiguity and lack of explicit structure in tasks. In this paper, we propose JsonTuning, a novel structure-to-structure approach for instruction tuning. By leveraging the versatility and structured nature of JSON to represent tasks, JsonTuning enhances generalization by helping the model understand essential task elements and their relations, improves robustness by minimizing ambiguity, and increases controllability by providing explicit control over the output. We conduct a comprehensive comparative study with diverse language models and evaluation benchmarks. Experimental results show that JsonTuning outperforms TextTuning in various applications, showcasing improved performance, adaptability, robustness, and controllability. By overcoming the limitations of TextTuning, JsonTuning demonstrates significant potential for more effective and reliable LLMs capable of handling diverse scenarios[1].

## 1 Introduction

Natural language processing has advanced significantly with large language models (LLMs) such as GPT-3 (Brown et al., 2020), PaLM (Chowdhery et al., 2022), and LLaMA (Touvron et al., 2023a), which excel in various tasks such as machine translation and sentiment analysis. However, effectively responding to human instructions remains challenging. Instruction tuning (Wei et al., 2022) addresses this by fine-tuning LLMs using explicit task instructions, improving task comprehension and execution. This approach has led to the success of instruction-following LLMs, such as InstructGPT (Ouyang et al., 2022) and ChatGPT (OpenAI, 2023a), in a wide range of applications.

Existing instruction tuning methods formulate all tasks as natural language generation (Wei et al., 2022; Sanh et al., 2022; Wang et al., 2022b; Chung et al., 2022). They describe the task and desired output with natural language, which is straightforward as LLMs are typically trained with the language modeling task. However, natural language instructions can sometimes be ambiguous or open to interpretation, leading to suboptimal understanding or unintended outputs from the model. Providing long and detailed instructions may be necessary for complex tasks, but it can also be cumbersome and challenging for users, thus reducing user satisfaction. Specifically, such text-to-text instruction tuning (TextTuning) methods suffer from the following limitations: (1) **Generalization**. TextTuning methods mix task elements and instructions in natural language texts, which can obscure the structure in tasks. This lack of explicit representation in the data structure may cause ambiguity and make it difficult for the model to understand the essential elements of tasks and their relations. Consequently, the model might struggle to learn and generalize intricate relations and dependencies present within the data, potentially hindering its overall performance and adaptability. (2) **Robustness**. Ambiguity in natural language texts can lead to models being more sensitive to variations in the input, resulting in less robust performance. TextTuning methods have been shown sensitive to phrasings of instructions (Sanh et al., 2022; Sun et al., 2023), variations of labels (Ye et al., 2023; Wei et al., 2023), and the order of options (Pezeshkpour & Hruschka, 2023). (3) **Controllability**.

---

[1]Code and model will be publicly available.

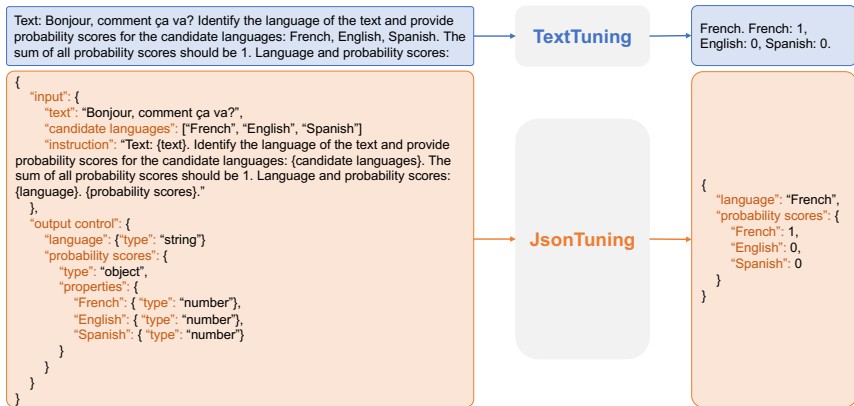

Figure 1: Overview of the typical TextTuning method and our proposed JsonTuning paradigm.

It can be difficult to provide a clear description or enforce a specific structure or format for the desired output due to the ambiguity of natural language (Han et al., 2023), preventing the model from effectively controlling the output.

To address the above limitations, it is crucial to incorporate explicit structure into the input and output representations during the instruction tuning process. Structured data representations such as JavaScript Object Notation (JSON) can help minimize misunderstandings and ensure a clearer communication of the intended task. Inspired by this, we propose a novel structure-to-structure approach called JsonTuning by leveraging the versatility and structured nature of JSON for instruction tuning. JsonTuning represents the inputs and outputs of all tasks as JSON structures, with the input JSON structure containing task input elements, instructions, and control information, and the output JSON structure encompassing task output elements. As depicted in Figure 1, JsonTuning addresses the limitations of TextTuning in the following ways: (1) **Generalization**. By explicitly representing the structure in tasks, JsonTuning helps the model understand essential elements of tasks and their underlying relations and ensures a consistent representation of data across different tasks, leading to improved generalization and adaptability to new tasks. (2) **Robustness**. JsonTuning helps minimize ambiguity and manage inconsistencies in the data, facilitating the model to process and generate accurate outputs when faced with input variations, resulting in enhanced robustness. (3) **Controllability**. JsonTuning offers explicit control over the output structure and content, enabling the model to effectively manage output generation. For the language detection task in Figure 1, JsonTuning clearly describes the output structure, including the organization and data types of output elements, which is challenging or even impossible to achieve using natural language texts alone.

We conduct a comparative study to demonstrate the advantages of JsonTuning by instruction-tuning five prominent pre-trained language models, namely LLaMA-7B, LLaMA-13B (Touvron et al., 2023a), LLaMA2-7B, LLaMA2-13B (Touvron et al., 2023b), and Falcon-7B (Penedo et al., 2023). The fine-tuning process involves a subset of the Flan 2022 collection (Chung et al., 2022) and structured tasks from InstructUIE (Wang et al., 2023a). Subsequently, we assess the performance of JsonTuning and TextTuning models in terms of generalization, robustness, and controllability across a diverse range of tasks, such as MMLU (Hendrycks et al., 2021), BBH (Suzgun et al., 2023), and tasks with intricate input and output structures. The experimental results reveal the following key findings: (1) JsonTuning outperforms TextTuning in terms of generalization across all language models and tasks. (2) JsonTuning models exhibit significantly greater robustness compared to TextTuning models with respect to variations in instructions and labels. (3) JsonTuning models demonstrate the ability to generalize to more complex structures even when trained on a limited number of simpler structured tasks and generate the desired output in a well-defined structured format.

## 2 JSONTUNING: STRUCTURE-TO-STRUCTURE INSTRUCTION TUNING

### 2.1 UNIFIED STRUCTURE-TO-STRUCTURE FORMULATION

We formulate instruction tuning as a structure-to-structure generation problem, representing task inputs and outputs using JSON format. Given a task $T$, we denote its input elements as $T_I =$

$(I_1, I_2, \ldots, I_n)$ and output elements as $T_O = (O_1, O_2, \ldots, O_m)$, where $I_i$ is the $i$th input element, and $O_i$ is the $i$th output element. Taking the multiple-choice question answering (MCQA) task in Table 1 for illustration, it has two input elements: *question* and *options* and an output element: *answer*. With $T_I$, $T_O$, the task prompt $TP$, label space $L$, and control information $C$, we construct the input JSON structure $S_I$ and output JSON structure $S_O$ as follows:

$$S_I = \{\text{``input''} : \{I_1 : v_1, I_2 : v_2, \ldots, I_n : v_n, l\_key : L, \text{``instruction''} : TP\}, \text{``output control''} : C\}$$

$$S_O = \{O_1 : u_1, O_2 : u_2, \ldots, O_m : u_m\}$$

where $v_i$ is the value of $I_i$, $u_i$ is the value of $O_i$, and $l\_key$ is the key in $S_I$ to indicate $L$ and varies for different tasks. For instance, $l\_key$ is *candiate answers* for the MCQA task since its output element is *answer*. We identify the following components for effective instruction tuning:

- **Task Prompt** $TP$. The task prompt $TP$ provides instructions for generating $T_O$ conditioned on $T_I$ and is necessary for instruction tuning. We incorporate a key named *instruction* in $S_I$ to provide such information.

- **Label Space** $L$. The label space $L$ is applicable only for tasks with limited label spaces and includes all possible outputs. For example, $L$ for the MCQA task comprises all candidate answers. Including $L$ in $S_I$ offers the following benefits: (1) Improving training consistency. For example, in the case of the MCQA task, if the correct option is *(A) Sundar Pichai*, the answer can be *A*, *(A)*, or the entire option *(A) Sundar Pichai*. These answers are all valid but may cause inconsistency during training since different datasets may use different types of answers. Moreover, there is a diverse range of tasks with limited label spaces. Incorporating $L$ unifies all these tasks and scenarios as a selection task, which involves choosing an item from the label space as the output. (2) Controlling the output. With $L$, we can restrict the output within a predetermined range.

- **Control Information** $C$. To improve the controllability of JsonTuning models, we incorporate the control information $C$ for each task, which specifies the structured format, explanations, and constraints for the output. In particular, we introduce a key called *output control* in $S_I$ to represent the control information. We employ JSON Schema to define control information, resulting in $C$ being a JSON structure as well. As shown in Table 1, for the named entity recognition (NER) task, its control information describes that its output is an array of objects, where each object has two properties: entity category and entity span, both of which are strings. Also, for the MCQA task, $C$ indicates that the answer is a string and should be one of the candidate answers, i.e., its label space. JSON Schema enables $C$ to clearly define the desired output.

With $S_I$ and $S_O$, we can employ a language model $M : S_I \rightarrow S_O$ for training and inference.

## 2.2 TUNING DATA

The Flan 2022 collection (Chung et al., 2022; Longpre et al., 2023) is a comprehensive and widely-used public instruction tuning collection consisting of over 1800 tasks. It integrates resources from Flan 2021 (Wei et al., 2022), P3++ (Sanh et al., 2022), Super-Natural Instructions (Wang et al., 2022b), and additional reasoning, dialogue, and program synthesis datasets. For our primary experiments, we randomly sample a subset from the Flan 2022 collection, maintaining the original collection's data proportion to ensure task diversity.

Despite the diverse tasks in the Flan 2022 collection, the input and output structures are relatively simple. The outputs for nearly all tasks are purely textual, lacking arrays, objects, or their combinations. Consequently, language models tuned with the Flan 2022 collection may struggle to generalize to diverse and complex structured tasks. To address this limitation, we introduce structured tasks for instruction tuning. Specifically, we employ information extraction (IE) tasks from InstructUIE (Wang et al., 2023a) as structured tasks for the following reasons: (1) they are well-defined and representative, as numerous structure prediction tasks, such as semantic role labeling and coreference resolution, can be formulated as IE tasks (Paolini et al., 2021; Wang et al., 2022a); (2) they possess complex input and output structures; (3) different IE task datasets have varying schemas, such as different entity categories and relations, thus fostering diversity. InstructUIE comprises three tasks: named entity recognition (NER), relation extraction (RE), and event extraction (EE). We utilize the NER and RE tasks for training, reserving the EE task for evaluation. Since the output structure of the EE task is more intricate than that of the NER and RE tasks, we can assess the instruction-tuned language models' capability to generalize to more complex structures. To encourage diversity, we

Table 1: Examples of TextTuning and JsonTuning. Each example is associated with a prompt consisting of an input template and an output template. We highlight the input template in brown and the output template in orange. MCQA refers to multiple-choice question answering, and NER represents named entity recognition.

| Method | Input | Output |
|---|---|---|
| **MCQA:** [Answering the following question: {question} {options}. Answer:, {answer}] | | |
| Text | Answering the following question: Who is the CEO of Google? (A) Sundar Pichai (B) Bill Gates (C) Tim Cook (D) Satya Nadella. Answer: | (A) |
| Json | {"input": { "question": "Who is the CEO of Google?", "options": "(A) Sundar Pichai (B) Bill Gates (C) Tim Cook (D) Satya Nadella", "candidate answers": ["(A)", "(B)", "(C)", "(D)"], "instruction": "Answering the following question: {question} {options}. Answer: {answer}" }, "output control": { "answer": { "type": "string", "description": "The answer should be one of the candidate answers in the input." } } } | {"answer": "(A)" } |
| **NER:** [definition: {definition}\ntext: {text}entity categories: {entity categories}\nentities:, {entities}] | | |
| Text | definition: Given a text and entity categories, your task is to scan the text and identify a list of named entities in it. Each entity contains an entity category and an entity span.\ntext: Tokyo is the capital of Japan.\nentity categories: location, person, organization\nentities: | [[location, Tokyo], [location, Japan]] |
| Json | { "input": { "definition": "Given a text and entity categories, your task is to scan the text and identify a list of named entities in it. Each entity contains an entity category and an entity span.", "text": "Tokyo is the capital of Japan.", "instruction": "definition: {definition}\ntext: {text}entity categories: {entity categories}\nentities: {entities}" }, "output control": { "entities": { "type": "array", "items": { "type": "object", "properties": { "entity category": {"type": "string"}, "entity span": {"type": "string"} } } } } } | {"entities": [ {entity category: location, entity span: Tokyo}, {entity category: location, entity span: Japan}] } |

uniformly select examples from the training sets of multiple datasets of each task for tuning. Further details regarding the training datasets of IE tasks can be found in Appendix A.

## 2.3 DATA PROCESSING

We use the defined data structures $S_I$ and $S_O$ in Section 2.1 to represent all tuning data in JSON structured format with the following data types: `object`, `array`, and `string`. The `number` and `boolean` types can be represented as the `string` type for simplicity. Further details regarding JSON and its utilization in data processing are available in Appendix C.

Following the approach in (Chung et al., 2022; Sanh et al., 2022; Wei et al., 2022), we employ multiple prompts for each task during instruction tuning, where each prompt $TP$ consists of an input template and an output template. For example, in the case of an MCQA prompt, the input template could be *"Answer the following question: {question} {options}. Answer:"*, and the output template could be *"{answer}"*. The prompt clearly indicates the essential task elements, namely *question*, *options*, and *answer*, as well as their relations. The tasks in the Flan 2022 collection already have multiple prompts. We manually construct 10 prompts each for NER and RE tasks for training, which can be found in Appendix D. Many tasks with limited label spaces in the Flan 2022 collection already include the label space in the task data source. For those that do not provide such information, we collect all possible task outputs in the data source to construct the label space. As for the control information $C$, all output elements of tasks in the Flan 2022 collection are of the `string` type, and we manually define $C$ for IE tasks, which can be found in Appendix E.

## 3 EXPERIMENTS

### 3.1 EXPERIMENTAL SETUP

**Pre-trained Language Models** We adopt five strong and prevalent pre-trained language models, namely LLaMA-7B, LLaMA-13B (Touvron et al., 2023a), LLaMA2-7B, LLaMA2-13B (Touvron et al., 2023b), and Falcon-7B (Penedo et al., 2023), for our experiments. These models are trained on trillions of tokens and are among the most widely used open-source language models.

**Evaluation Tasks and Datasets** We focus on performance on unseen datasets and tasks. We evaluate models on popular aggregated benchmarks: MMLU (Hendrycks et al., 2021) consisting of 57 tasks of exam questions and BBH (Suzgun et al., 2023) including 23 challenging tasks from BIG-Bench (Srivastava et al., 2023) following Chung et al. (2022). In addition, we adopt tasks with complex input and output structures for evaluation. Specifically, we use the NER, RE, and EE tasks from InstructUIE (Wang et al., 2023a), the table question answering (TQA) task, which involves answering questions based on a given structured table, and the NL2SQL task, requiring the conversion of natural language queries into SQL using a provided database schema. For NER and RE, we use datasets unseen during training. Specifically, we use 5 datasets, namely, AI, literature, music, politics, and science, from CrossNER (Liu et al., 2021) for the NER task and 2 datasets, namely CoNLL2004 (Roth & Yih, 2004) and FewRel (Han et al., 2018), for the RE task. For the unseen EE task, we use ACE2005 (Walker et al., 2006), CASIE (Satyapanich et al., 2020), and PHEE (Sun et al., 2022) datasets for evaluation. We use the WikiTableQuestions (Pasupat & Liang, 2015) dataset for the TQA task and the Spider (Yu et al., 2018) dataset for the NL2SQL task. Apart from datasets in MMLU and BBH, we randomly select up to 500 examples for each dataset from its test set[2] for evaluation so that a single dataset will not dominate the results of its task and the evaluation cost is acceptable. The details of evaluation datasets and prompts are in Appendix E.

**Evaluation Metrics** We use accuracy for MMLU and BBH following Chung et al. (2022), entity F1 for the NER task, relation boundary F1 for the RE task, event trigger F1 and argument F1 for the EE task following Wang et al. (2023a), accuracy for the TQA task following Pasupat & Liang (2015), and execution accuracy for the NL2SQL task following Yu et al. (2018).

**Implementation Details** We employ the parameter-efficient method LoRA (Low Rank Adaptation) (Hu et al., 2022) for fine-tuning. The rank is set to 8. We use 50K examples from the Flan collection 2022 and 10K examples from structured tasks in InstructUIE, with an equal division between the NER and RE tasks, and train the learnable parameters for 3 epochs with a batch size of 64. For model optimization, we use the AdamW (Loshchilov & Hutter, 2019) optimizer with linear learning rate decay, and the peak learning rate is set to 1e-3. We set the maximum length as 2048 for training and evaluation.

## 3.2 GENERALIZATION RESULTS

Table 2 presents the zero-shot generalization results of JsonTuning and TextTuning using five distinct language models. We have the following observations:

- *JsonTuning surpasses TextTuning in the majority of tasks and models.* This is evident from the higher average scores for JsonTuning across all models and tasks, where JsonTuning achieves an overall average score of 25.54 compared to TextTuning's 22.24. This suggests that JsonTuning is a more effective method for instruction tuning.

- *JsonTuning significantly improves the model's ability to tackle complex structured tasks.* Json-Tuning models consistently outperform TextTuning approaches on tasks with complex structures, such as NER, EE, and NL2SQL. Models trained with JsonTuning can adapt to intricate EE structures, even when only trained on simpler NER and RE structures. In contrast, TextTuning methods rarely generate valid EE structures, highlighting the superior controllability of JsonTuning. Notably, simply improving pre-training and enlarging the model size may not result in consistent performance enhancements for these tasks. For instance, transitioning from the LLaMA-7B to the better pre-trained LLaMA2-7B model does not improve generalization results on the EE task, and increasing the model size for LLaMA from 7B to 13B does not enhance TextTuning performance on the NER task. These observations suggest that JsonTuning is the preferred choice for instruction tuning in tasks requiring the process and prediction of complex structures.

- *Pre-training is vital for improving generalization, and JsonTuning appears to be more beneficial for models with limited capabilities.* The table shows that Falcon considerably underperforms LLaMA and LLaMA2 across all tasks, indicating the impact of advanced pre-training on generalization. Interestingly, Falcon-7B with JsonTuning exhibits a substantial improvement over

---

[2]We use the development set for Spider since its test set is not publicly available.

Table 2: Generalization results.

| Model | Method | MMLU | BBH | NER | RE | EE | TQA | NL2SQL | Average |
|-------|--------|------|-----|-----|-----|-----|-----|--------|---------|
| Falcon-7B | Text | 24.64 | 20.64 | 16.92 | 4.58 | 0.16 / 0.00 | 3.00 | 2.00 | 10.27 |
|  | Json | 34.13 | 32.61 | 21.92 | 6.94 | 0.00 / 0.31 | 3.60 | 1.40 | **14.39** |
| LLaMA-7B | Text | 43.11 | 32.48 | 36.47 | 13.60 | 1.08 / 0.00 | 18.20 | 8.60 | 21.86 |
|  | Json | 44.69 | 37.08 | 41.80 | 15.56 | 3.09 / 8.24 | 16.40 | 16.40 | **25.37** |
| LLaMA-13B | Text | 49.49 | 39.07 | 36.45 | 20.19 | 1.59 / 0.00 | 14.80 | 17.80 | 25.51 |
|  | Json | 48.98 | 40.47 | 42.81 | 22.61 | 3.98 / 11.09 | 14.40 | 21.40 | **28.22** |
| LLaMA2-7B | Text | 46.36 | 37.89 | 38.82 | 20.42 | 0.42 / 0.00 | 16.60 | 10.80 | 24.44 |
|  | Json | 47.95 | 39.23 | 43.68 | 24.71 | 4.00 / 4.16 | 20.00 | 11.20 | **27.26** |
| LLaMA2-13B | Text | 52.30 | 41.91 | 39.45 | 22.65 | 1.40 / 0.00 | 23.80 | 23.20 | 29.14 |
|  | Json | 51.88 | 42.85 | 46.70 | 22.76 | 6.82 / 10.40 | 27.40 | 26.40 | **32.37** |
| Average | Text | 43.18 | 34.40 | 33.62 | 16.29 | 0.93 / 0.00 | 15.28 | 12.48 | 22.24 |
|  | Json | **45.53** | **38.50** | **39.38** | **18.52** | **3.58 / 6.84** | **16.36** | **15.36** | **25.54** |

TextTuning on tasks such as MMLU and BBH, emphasizing the importance of JsonTuning, especially when working with less capable models. This suggests that JsonTuning enables the model to more effectively utilize its capabilities and knowledge in responding to human instructions.

## 3.3 ROBUSTNESS RESULTS

The robustness of instruction-tuned language models is of paramount importance for their successful deployment across a diverse range of tasks. In this section, we assess the model's resilience against varying prompts and unseen labels, which have been identified as challenging aspects for instruction-tuned models in prior research (Sanh et al., 2022; Sun et al., 2023; Ye et al., 2023).

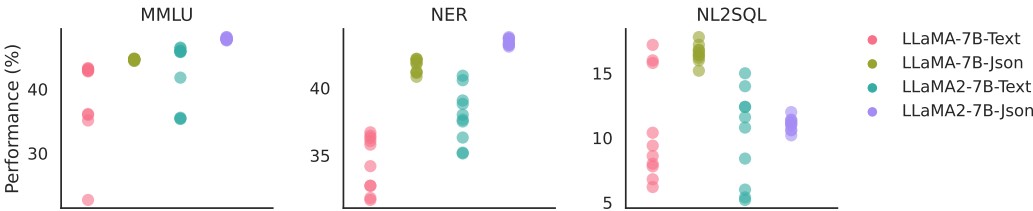

Figure 2: Performance of JsonTuning and TextTuing models with different prompts.

To evaluate prompt robustness, we employ 10 distinct prompts for the MMLU benchmark, the NER task, and the NL2SQL task. Detailed information can be found in Appendix E. Figure 2 illustrates the performance of JsonTuning and TextTuning models on these tasks when subjected to different prompts. Our findings reveal that JsonTuning models exhibit greater robustness compared to TextTuning models, as evidenced by higher mean performance and reduced variance. This can be attributed to JsonTuning's ability to effectively differentiate between instructions and task elements, thereby minimizing ambiguity and enhancing robustness.

In terms of label robustness, we substitute the MMLU label space with previously unseen label spaces. The original label space for MMLU is {(A), (B), (C), (D)}, with these option letters frequently encountered in the training dataset. We replace this label space with two alternatives: {(W), (X), (Y), (Z)} and {($), (€), (£), (¥)}, denoted by Unseen-1 and Unseen-2, respectively. These label spaces were not present during instruction tuning. As shown in Figure 3, JsonTuning models exhibit a smaller performance decline and consistently outperform TextTuning models in all scenarios. This can be ascribed to JsonTuning's enhanced consistency for tasks with constrained label spaces and its diminished ambiguity, which facilitates easier understanding.

Table 3: Case studies focusing on controllability. Each example displays its input along with the model's prediction.

| Method | Input | Prediction |
|---|---|---|
| **Task: Language detection with probability scores** | | |
| Text | Text: Bonjour, comment ça va? Identify the language of the text and provide probability scores for the candidate languages: French, English, Spanish. The sum of all probability scores should be 1. Language and probability scores: | French .7 |
| Json | { "input": { "text": "Bonjour, comment ça va?", "candidate languages": ["French", "English", "Spanish"], "instruction": "Text: {text}. Identify the language of the text and provide probability scores for the candidate languages: {candidate languages}. The sum of all probability scores should be 1. Language and probability scores: {language} {probability scores}" }, "output control": { "language": {"type": "string"}, "probability scores": { "type": "object", "properties": { "French": {"type": "string" }, "English": {"type": "string" }, "Spanish": {"type": "string" } } } } } | {"language": "French", "probability scores": { "French": 0.98, "English": 0.01, "Spanish": 0.01} } |
| **Task: Joke generation with humor style** | | |
| Text | Generate the joke with a specific humor style (e.g., pun, sarcasm): | What do you call a deer with no eyes? No idea. |
| Json | { "input": { "instruction": "Generate the joke with a specific humor style (e.g., pun, sarcasm): {humor style} {joke}" }, "output control": { "humor style": {"type": "string"}, "joke": {"type": "string"} } } | {"humor style": "pun", "joke": "What did the pirate say when he was given a piece of paper?"} |
| **Task: Intent detection and slot filling** | | |
| Text | Text: Set an alarm for 7 AM tomorrow. Detect the intent of the text and extract time and date slots from the text: | Set an alarm for 7 AM tomorrow |
| Json | { "input": { "text": "Set an alarm for 7 AM tomorrow.", "instruction": "Text: {text}. Detect the intent or purpose of the text and extract time and date slots from the text:" }, "output control": { "intent": {"type": "string"}, "slots": { "type": "object", "properties": { "time": {"type": "string"}, "date": {"type": "string"} } } } } | {"intent": "setAlarm", "slots": {"time": "7:00", "date": "tomorrow"} } |

## 3.4 CASE STUDIES ON CONTROLLABILITY

In previous sections, we have demonstrated that JsonTuning models possess the capacity to control the output and generalize across complex structures. In this section, we present case studies to qualitatively illustrate the controllability of JsonTuning models. For this purpose, we utilize LLaMA2-13B trained with both JsonTuning and TextTuning approaches.

As evidenced by Table 3, JsonTuning effectively enables the model to identify the desired output, generating results in a well-structured format. In contrast, the TextTuning model fails to adequately adhere to the provided instructions. For example, in the language detection task, the TextTuning model struggles to provide

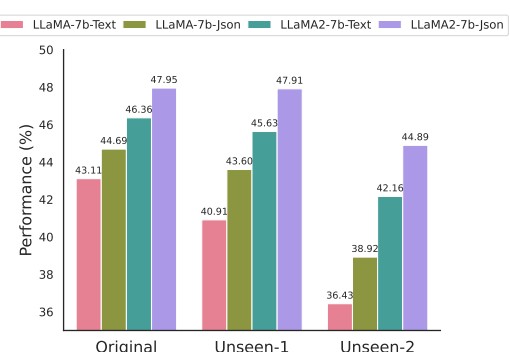

Figure 3: Performance of JsonTuning and TextTuning models with different label spaces on MMLU.

clear probability scores. The output, such as ".7", is ambiguous and difficult to interpret. By comparison, JsonTuning successfully follows the instruction, delivering scores that meet the specified requirements. Additional case studies can be found in Appendix B. It is crucial to emphasize that controllability is a vital aspect when deploying language models for real-world applications. JsonTuning, in comparison to TextTuning, offers a significantly improved method for achieving this.

Table 4: Ablation results for LLaMA-7B concerning label space and control information. Tasks marked with † are seen during training and evaluated with unseen datasets.

| Method | MMLU | BBH | NER† | RE† | EE | TQA | NL2SQL | Average |
|---|---|---|---|---|---|---|---|---|
| JsonTuning | 44.69 | 37.08 | 41.80 | 15.56 | 3.09 / 8.24 | 16.40 | 16.40 | **25.37** |
| w/o label space | 43.04 | 34.93 | 42.25 | 18.66 | 4.27 / 1.99 | 17.80 | 12.40 | 24.60 |
| w/o control information | 43.94 | 35.42 | 46.77 | 16.70 | 1.04 / 0.26 | 14.40 | 12.40 | 24.33 |
| w/o both | 42.66 | 36.23 | 47.65 | 16.23 | 0.99 / 0.29 | 10.40 | 12.80 | 23.80 |

## 4 ANALYSIS

In this section, we investigate the performance of LLaMA-7B employing JsonTuning and Text-Tuning under various conditions, including different method designs, data sizes, and numbers of structured task examples.

**Are label space and control information critical for JsonTuning?** From Table 4, we can make the following observations: (1) Generally, eliminating label space or control information does not negatively impact the model's performance on seen tasks such as NER and RE; in fact, the performance on these tasks may even improve. However, doing so does hinder performance on unseen tasks, resulting in a lower average performance. Removing both elements further diminishes the performance, indicating that they aid the model in generalizing to unseen tasks rather than overfitting to seen tasks. (2) Excluding label space has a more pronounced effect on the model's performance on MMLU, as all tasks within it have a limited label space. This finding suggests that incorporating label space during training and evaluation is beneficial for tasks with restricted label spaces. (3) The model's ability to generalize to more complex structures is substantially reduced without control information. This is evident by the considerable performance drop on the EE task, demonstrating that control information is essential for the model to generalize effectively to complicated structures.

**What are the effects of different data sizes on generalization?** In the primary experiments, we utilize a total of 60K data points, comprising 50K from the Flan 2022 collection and 10K from structured tasks for tuning. In this analysis, we alter the data size while maintaining their relative ratio to examine the effects of different data sizes on generalization. Specifically, we train LLaMA-7B with four different data sizes: 12K, 36K, 60K, and 120K, and evaluate the models on the MMLU benchmark, the NER task, and the NL2SQL task. Figure 4 reveals the following observations: (1) LLaMA-7b-Json consistently outperforms LLaMA-7b-Text across all tasks and data sizes, indicating the superior generalization capabilities of the JsonTuning model. (2) Increasing the data size for instruction tuning does not necessarily result in performance improvement, suggesting that enlarging the data size may not be an effective approach to enhance the model's generalization abilities.

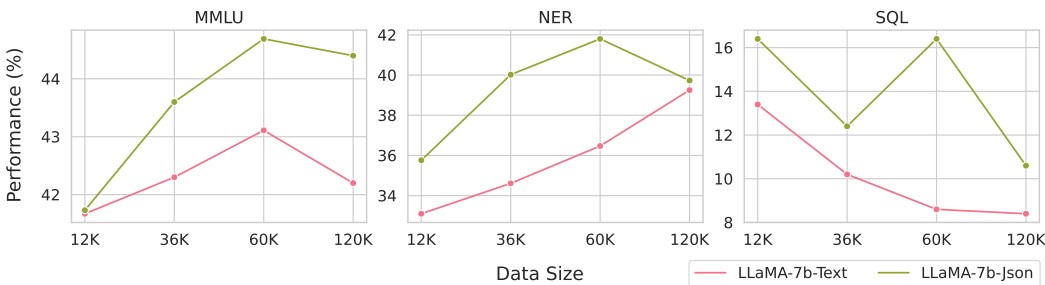

Figure 4: Performance of LLaMA-7B trained using JsonTuning and TextTuning across varying data sizes.

**Are structured tasks essential for instruction tuning?** To investigate this, we keep the number of examples from the Flan 2022 collection constant and vary the number of examples from structured tasks. Specifically, we use 50K data points from the Flan 2022 collection and 0K, 2K, 6K, 10K, and 20K data points from IE tasks for the experiments. Figure 5 reveals the following insights:

(1) Incorporating structured tasks for training may not enhance the model's generalization ability on tasks without complex structures. Introducing structured tasks for tuning does not improve the model's performance on MMLU, a benchmark without intricate input and output structures. (2) Structured tasks significantly impact the model's generalization performance on tasks with complex output structures. Without structured tasks for training, the model's performance on the NER task is 0 for both JsonTuning and TextTuning. However, the performance significantly improves when introducing only 2K data points from structured tasks for training. This highlights the importance of structured tasks for instruction-tuned models to generalize to tasks with complex output structures. (3) Structured tasks have a milder impact on the model's generalization performance on tasks with complex input structures. Introducing an appropriate number of structured tasks can enhance the model's performance on the NL2SQL task, which requires processing a structured database schema. This suggests that training the model with structured tasks aids in processing and understanding complex structures. In summary, the decision to use structured tasks for instruction tuning depends on the application scenarios. However, regardless of the scenario, JsonTuning consistently appears to be a superior method for instruction tuning compared to TextTuning.

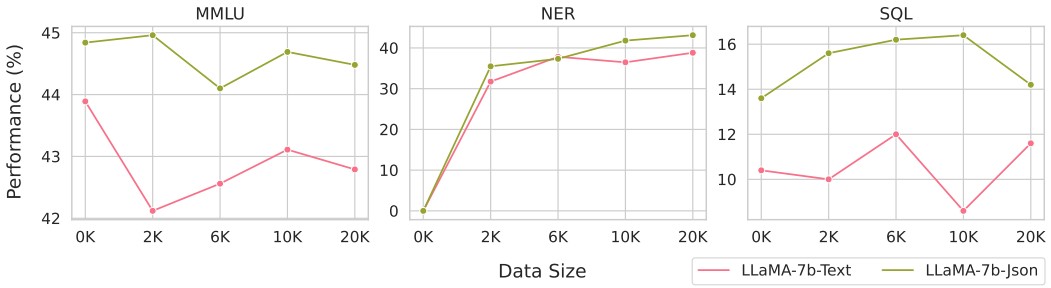

Figure 5: Performance of LLaMA-7B trained using JsonTuning and TextTuning with different numbers of examples of structured tasks.

## 5 RELATED WORK

The development of large language models (LLMs) has had a profound impact on the AI community, with models such as ChatGPT (OpenAI, 2023a) and GPT-4 (OpenAI, 2023b) driving discussions on the potential of artificial general intelligence (AGI) and redefining the boundaries of what AI systems can achieve. These advancements have also led to a surge in the development and release of open-source LLMs (Zhang et al., 2022; Team, 2023; Penedo et al., 2023; Touvron et al., 2023a;b), fostering innovation and collaboration within the research community.

Instruction tuning (Wei et al., 2022; Mishra et al., 2022; Sanh et al., 2022; Chung et al., 2022; Wang et al., 2023b; Taori et al., 2023; Liu et al., 2023) has emerged as a promising research direction that leverages LLM capabilities to enhance responsiveness to human instructions. Collections such as Super-NaturalInstructions (Wang et al., 2022b), OPT-IML Bench (Iyer et al., 2022), and the Flan 2022 collection (Chung et al., 2022) have accelerated the development of instruction-tuned models. To advance instruction tuning, researchers have explored learning from human feedback (Stiennon et al., 2020; Ouyang et al., 2022; Bai et al., 2022; Scheurer et al., 2023), automatic data generation (Wang et al., 2023b; Peng et al., 2023; Xu et al., 2023), and data selection (Zhou et al., 2023; Cao et al., 2023; Lu et al., 2023). Our JsonTuning approach offers an alternative perspective on data representation to enhance instruction tuning in terms of generalization, robustness, and controllability.

## 6 CONCLUSION

This paper introduces JsonTuning, a novel approach designed to overcome the limitations of conventional text-to-text instruction tuning methods for language models. By utilizing the structured data format of JSON for explicit task representation, JsonTuning significantly improves the model's generalization, robustness, and controllability. Our experimental results and case studies highlight the benefits of JsonTuning in generalizing to unseen tasks and datasets, maintaining robustness against varying prompts and label spaces, and demonstrating controllability in diverse scenarios.

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

## A  DATASETS OF INFORMATION EXTRACTION TASKS

Table 5 reports the training and evaluation datasets of information extraction tasks.

Table 5: Information Extraction (IE) datasets utilized for training and evaluation. |Schema| represents the number of entity categories in the named entity recognition (NER) task, the number of relations in the relation extraction (RE) task, and the number of event categories (outside the parenthesis) along with the number of argument categories (inside the parenthesis) in the event extraction (EE) task.

| Task | Dataset | \|Schema\| |
|------|---------|---------|
| | **Training** | |
| NER | ACE2004 (Walker et al., 2006) | 7 |
| | ACE2005 (Walker et al., 2006) | 7 |
| | broad_twitter_corpus (Derczynski et al., 2016) | 3 |
| | CoNLL2003 (Tjong Kim Sang & De Meulder, 2003) | 4 |
| | multiNERD (Tedeschi & Navigli, 2022) | 16 |
| | Ontonotes (Hovy et al., 2006) | 18 |
| | polyglot-NER (Al-Rfou et al., 2015) | 3 |
| | tweetNER7 (Ushio et al., 2022) | 7 |
| | wikiann (Pan et al., 2017) | 3 |
| | wikineural (Tedeschi et al., 2021) | 3 |
| | AnatEM (Pyysalo & Ananiadou, 2013) | 1 |
| | bc2gm (Kocaman & Talby, 2021) | 1 |
| | bc4chemd (Krallinger et al., 2015) | 1 |
| | bc5cd (Li et al., 2016) | 2 |
| | FabNER (Kumar & Starly, 2021) | 12 |
| | FindVehicle (Guan et al., 2023) | 21 |
| | HarveyNER (Chen et al., 2022) | 4 |
| | ncbi-disease (Dogan et al., 2014) | 1 |
| RE | GIDS (Nayak et al., 2021) | 4 |
| | kbp37 (Zhang & Wang, 2015) | 18 |
| | NYT (Riedel et al., 2010) | 24 |
| | NYT11 HRL (Takanobu et al., 2019) | 12 |
| | SciERC (Luan et al., 2018) | 7 |
| | semeval RE (Hendrickx et al., 2010) | 10 |
| | **Evaluation** | |
| NER | CrossNER_AI (Liu et al., 2021) | 14 |
| | CrossNER_literature (Liu et al., 2021) | 12 |
| | CrossNER_music (Liu et al., 2021) | 13 |
| | CrossNER_politics (Liu et al., 2021) | 9 |
| | CrossNER_science (Liu et al., 2021) | 17 |
| RE | CoNLL2004 (Roth & Yih, 2004) | 5 |
| | FewRel (Han et al., 2018) | 5 |
| EE | ACE2005 (Walker et al., 2006) | 33(22) |
| | CASIE (Satyapanich et al., 2020) | 5(26) |
| | PHEE (Sun et al., 2022) | 2(16) |

## B  ADDITIONAL CASE STUDIES ON CONTROLLABILITY

Table 6 presents additional case studies on controllability.

Table 6: Case studies focusing on controllability. Each example displays its input along with the model's prediction.

| Method | Input | Prediction |
|---|---|---|
| **Task: Translate a sentence to multiple languages** | | |
| Text | Sentence: How are you? Translate the sentence to multiple languages: French and Spanish. Translations: | Comment allez-vous? |
| Json | {"input": { "sentence": "How are you?", "instruction": "Sentence: {sentence}. Translate the sentence to two languages: French and Spanish. Translations: {translations}" }, "output control": { "translations": { "type": "object", "properties": { "French": { "type": "string", "description": "The French translation of the sentence." }, "Spanish": { "type": "string", "description": "The Spanish translation of the sentence." } } } } } | {"translations": {"French": "Comment allez-vous?", "Spanish": "Cómo estás?"} } |
| **Task: Sentiment analysis with emotion** | | |
| Text | Given the sentiment labels: positive, negative, and neutral, and text: I just won a lottery! Life is amazing!, provide the sentiment label and emotion associated with the text. Sentiment label and emotion: | positive |
| Json | { "input": { "text": "I just won a lottery! Life is amazing!", "sentiment labels": ["positive", "negative", "neutral"], "instruction": "Given the sentiment labels: {sentiment labels} and text: {text}, provide the sentiment label and emotion associated with the text. Sentiment label and emotion: {sentiment label} {emotion}" }, "output control": { "sentiment label": {"type": "string"}, "emotion": {"type": "string"} } } | {"sentiment label": "positive", "emotion": "joy"} |

## C  INTRODUCTION TO JSON AND ITS UTILIZATION

### C.1  JSON DATA TYPES AND SYNTAX

JSON data is represented using a combination of the following data types:

- **Object:** An unordered collection of key-value pairs, enclosed in curly braces {}. The keys are strings, and the values can be any of the JSON data types.
- **Array:** An ordered list of values, enclosed in square brackets []. The values can be any of the JSON data types.
- **String:** A sequence of Unicode characters, enclosed in double quotes.
- **Number:** A numeric value, which can be an integer or a floating-point number.
- **Boolean:** A value that is either true or false.
- **Null:** A special keyword denoting a null value.

In this paper, we focus on the object, array, and string types, as the number and boolean types can be represented as the string type for simplicity. By combining these simple data types, JSON can represent various structured data. This flexibility allows language models that understand basic data types to potentially generalize to more complex structures.

### C.2  JSON SCHEMA

JSON Schema employs a JSON-based format for defining the structure of JSON data, specifying properties like data types, required fields, and permissible values for JSON objects. It uses many keywords to define and validate JSON data. In this paper, we use the following keywords to construct the control information $C$:

- **type:** Specifies the data type of a JSON value, such as object, array, and string.
- **description:** Provides explanations and clarifications about the purpose and constraints of a specific element or property.

- **items:** Defines the elements of an array and their data types.
- **properties:** Describes the properties of an object, including their data types and constraints.

We may introduce more keywords to further improve the model's controllability in the future.

## C.3 JSON EXAMPLE

```
1  {
2      "type": "object",
3      "properties": {
4          "first name": { "type": "string" },
5          "last name": { "type": "string" },
6          "phone numbers": {
7              "type": "array",
8              "items": { "type": "string" }
9          }
10         "address": {
11             "type": "object",
12             "properties": {
13                 "city": { "type": "string" },
14                 "state": { "type": "string" },
15                 "country": { "type": "string" }
16             }
17         }
18     }
19 }
```

The example provided above employs JSON Schema to define a person object. This object comprises multiple properties, each with its own type. JSON's ability to handle nested structures allows it to support a wide range of complex and diverse structures. An instance of the person object can be seen below:

```
1  {
2      "first name": "John",
3      "last name": "Doe",
4      "phone numbers": ["12345", "678910"],
5      "address": {
6          "city": "AnyCity",
7          "state": "AnyState",
8          "country": "AnyCountry"
9      }
10 }
```

## D NAMED ENTITY RECOGNITION AND RELATION EXTRACTION PROMPTS

We create prompts for both the named entity recognition (NER) and relation extraction (RE) tasks. For the RE task, we develop two sets of prompts: one for datasets with entity categories and another for datasets without entity categories. Each prompt comprises an input template and an output template, which are highlighted in blue and orange, respectively.

- NER

Prompt 1: [definition: {definition}\ntext: {text}\nentity categories: {entity categories}\nentities:, {entities}]
Prompt 2: [definition: {definition}\nentity categories: {entity categories}\ntext: {text}\nentities:, {entities}]
Prompt 3: [text: {text}\ndefinition: {definition}\nentity categories: {entity categories}\nentities:, {entities}]
Prompt 4: [text: {text}\nentity categories: {entity categories}\ndefinition: {definition}\nentities:, {entities}]
Prompt 5: [entity categories: {entity categories}\ntext: {text}\ndefinition: {definition}\nentities:, {entities}]
Prompt 6: [entity categories: {entity categories}\ndefinition: {definition}\ntext: {text}\nentities:, {entities}]
Prompt 7: [{definition}\ntext: {text}\nentity categories: {entity categories}\nentities:, {entities}]
Prompt 8: [{definition}\nentity categories: {entity categories}\ntext: {text}\nentities:, {entities}]
Prompt 9: [text: {text}\nentity categories: {entity categories}\n{definition}\nentities:, {entities}]
Prompt 10: [entity categories: {entity categories}\ntext: {text}\n{definition}\nentities:, {entities}]

- RE (with entity categories)

Prompt 1: [definition: {definition}\ntext: {text}\nentity categories: {entity categories}\nrelations: {relations}\nrelational triplets:, {relational triplets}]
Prompt 2: [definition: {definition}\nentity categories: {entity categories}\nrelations: {relations}\ntext: {text}\nrelational triplets:, {relational triplets}]
Prompt 3: [definition: {definition}\ntext: {text}\nrelations: {relations}\nentity categories: {entity categories}\nrelational triplets:, {relational triplets}]
Prompt 4: [definition: {definition}\nrelations: {relations}\nentity categories: {entity categories}\ntext: {text}\nrelational triplets:, {relational triplets}]
Prompt 5: [text: {text}\ndefinition: {definition}\nentity categories: {entity categories}\nrelations: {relations}\nrelational triplets:, {relational triplets}]
Prompt 6: [text: {text}\nentity categories: {entity categories}\nrelations: {relations}\ndefinition: {definition}\nrelational triplets:, {relational triplets}]
Prompt 7: [text: {text}\ndefinition: {definition}\nrelations: {relations} \nentity categories: {entity categories}\nrelational triplets:, {relational triplets}]
Prompt 8: [text: {text}\nrelations: {relations}\nentity categories: {entity categories}\ndefinition: {definition}\nrelational triplets:, {relational triplets}]
Prompt 9: [entity categories: {entity categories}\nrelations: {relations}\ntext: {text}\ndefinition: {definition}\nrelational triplets:, {relational triplets}]
Prompt 10: [relations: {relations}\nentity categories: {entity categories}\ndefinition: {definition}\ntext: {text}\nrelational triplets:, {relational triplets}]

- RE (without entity categories)

> Prompt 1: [definition: {definition}\ntext: {text}\nrelations: {relations}\nrelational triplets:, {relational triplets}]
> Prompt 2: [definition: {definition}\nrelations: {relations}\ntext: {text}\nrelational triplets:, {relational triplets}]
> Prompt 3: [text: {text}\ndefinition: {definition}\nrelations: {relations}\nrelational triplets:, {relational triplets}]
> Prompt 4: [text: {text}\nrelations: {relations}\ndefinition: {definition}\nrelational triplets:, {relational triplets}]
> Prompt 5: [relations: {relations}\ntext: {text}\ndefinition: {definition}\nrelational triplets:, {relational triplets}]
> Prompt 6: [relations: {relations}\ndefinition: {definition}\ntext: {text}\nrelational triplets:, {relational triplets}]
> Prompt 7: [{definition}\ntext: {text}\nrelations: {relations}\nrelational triplets:, {relational triplets}]
> Prompt 8: [{definition}\nrelations: {relations}\ntext: {text}\nrelational triplets:, {relational triplets}]
> Prompt 9: [text: {text}\nrelations: {relations}\n{definition}\nrelational triplets:, {relational triplets}]
> Prompt 10: [relations: {relations}\ntext: {text}\n{definition}\nrelational triplets:, {relational triplets}]

## E  EVALUATION PROMPTS AND EXAMPLES

For MMLU and BBH, we utilize the prompts from the Flan2022 collection designed for question answering[3]. For other evaluation tasks, we create prompts based on their respective task components and definitions. Each prompt includes an input template and an output template, highlighted in blue and orange, respectively. Further details can be found in the subsequent sections.

### E.1  GENERALIZATION

All tasks employ a single prompt for evaluation, except for the RE task. The RE task utilizes two prompts: one for datasets with entity categories and another for datasets without entity categories. The prompts and examples are presented below:

---

[3]For more details, see `https://github.com/google-research/FLAN/blob/main/flan/v2/templates.py`.

- MMLU

Prompt: [{question}\n{options_}\nAnswer:, {answer}]

TextTuning Example:
Input: The following is a multiple choice question about global facts.\nControlling for inflation and PPP-adjustment, about how much did GDP per capita increase from 1950 to 2016 in Japan? Options:\n(A) by 5 fold\n(B) by 10 fold\n(C) by 15 fold\n(D) by 20 fold\nAnswer:
Output: (C)

JsonTuning Example:
Input: {"input": { "question": "The following is a multiple choice question about global facts.\nControlling for inflation and PPP-adjustment, about how much did GDP per capita increase from 1950 to 2016 in Japan?", "options_": "Options:\n(A) by 5 fold\n(B) by 10 fold\n(C) by 15 fold\n(D) by 20 fold", "candidate answers": ["(A)", "(B)", "(C)", "(D)"], "instruction": "{question}\n{options_}\nAnswer: {answer}" }, "output control": { "answer": { "type": "string", "description": "The answer should be one of the candidate answers in the input." } } }
Output: {"answer": "(C)"}

- BBH

Prompt: [Q: {question}\nA:, {answer}]

TextTuning Example:
Input: Q: ((-1 + 2 + 9 * 5) - (-2 + -4 + -4 * -7)) =\nA:
Output: 24

JsonTuning Example:
Input: {"input": { "question": "((-1 + 2 + 9 * 5) - (-2 + -4 + -4 * -7)) =", "instruction": "Q: {question}\nA: {answer}" }, "output control": { "answer": { "type": "string" } } }
Output: {"answer": "24"}

- NER

Prompt: [definition: {definition}\ntext: {text}\nentity categories: {entity categories}\nentities:, {entities}]

TextTuning Example:

Input: definition: Given a text and entity categories, your task is to scan the text and identify a list of named entities in it. Each entity contains an entity category and an entity span. An entity span refers to the specific portion of the text that represents an entity. An entity category refers to the category to which an entity belongs.\ntext: He also co-wrote Possible, which has been used as a theme song for the 2005 Southeast Asian Games.\nentity categories: location, event, country, band, person, song, musical artist, music genre, else, album, organization, award, musical instrument\nentities:

Output: [["song", "Possible"], ["event", "2005 Southeast Asian Games"]]

JsonTuning Example:

Input: {"input": { "definition": "Given a text and entity categories, your task is to scan the text and identify a list of named entities in it. Each entity contains an entity category and an entity span. An entity span refers to the specific portion of the text that represents an entity. An entity category refers to the category to which an entity belongs.", "text": "He also co-wrote Possible, which has been used as a theme song for the 2005 Southeast Asian Games.", "entity categories": [ "location", "event", "country", "band", "person", "song", "musical artist", "music genre", "else", "album", "organization", "award", "musical instrument" ], "instruction": "definition: {definition}\ntext: {text}\nentity categories: {entity categories}\nentities: {entities}", }, "output control": { "entities": { "type": "array", "items": { "type": "object", "properties": { "entity category": { "type": "string", "description": "The entity category should be one of the entity categories provided in the input." }, "entity span": { "type": "string", "description": "The entity span should be a continuous span in the text provided in the input." } } } } } }

Output: { "entities": [ { "entity category": "song", "entity span": "Possible" }, { "entity category": "event", "entity span": "2005 Southeast Asian Games" } ] }

- RE (with entity categories)

Prompt: [definition: {definition}\ntext: {text}\nentity categories: {entity categories}\nrelations: {relations}\nrelational triplets:, {relational triplets}]

TextTuning Example:
Input: definition: Given a text, entity categories, and relations, your goal is to scan the text and identify a list of relational triplets in it. Each relational triplet contains a head entity category, a head entity span, a relation, a tail entity category, and a tail entity span. The head entity is the subject from which the relation originates. The relation represents the specific relation between the head entity and the tail entity. The tail entity is the object which the relation points. An entity span refers to the specific portion of the text that represents an entity. An entity category refers to the category to which an entity belongs.\ntext: In 1822, the 19th president of the United States, Rutherford B. Hayes, was born in Delaware, Ohio. \nentity categories: Organization, Location, People\nrelations: Kill, Work for, Located in, Live in, Organization based in\nrelational triplets:
Output: [["People", "Rutherford B. Hayes", "Live in", "Location", "Delaware, Ohio"]]

JsonTuning Example:
Input: {"input": { "definition": "Given a text, entity categories, and relations, your goal is to scan the text and identify a list of relational triplets in it. Each relational triplet contains a head entity category, a head entity span, a relation, a tail entity category, and a tail entity span. The head entity is the subject from which the relation originates. The relation represents the specific relation between the head entity and the tail entity. The tail entity is the object which the relation points. An entity span refers to the specific portion of the text that represents an entity. An entity category refers to the category to which an entity belongs.", "text": "In 1822, the 19th president of the United States, Rutherford B. Hayes, was born in Delaware, Ohio.", "entity categories": [ "Organization", "Location", "People" ], "relations": [ "Kill", "Work for", "Located in", "Live in", "Organization based in" ], "instruction": "definition: {definition}\ntext: {text}\nentity categories: {entity categories}\nrelations: {relations}\nrelational triplets: {relational triplets}" }, "output control": { "relational triplets": "type": "array", "items": { "type": "object", "properties": { "head entity category": { "type": "string", "description": "The head entity category should be one of the entity categories provided in the input." }, "head entity span": { "type": "string", "description": "The head entity span should be a continuous span in the text provided in the input." }, "relation": { "type": "string", "description": "The relation should be one of the relations provided in the input." }, "tail entity category": { "type": "string", "description": "The tail entity category should be one of the entity categories provided in the input." }, "tail entity span": { "type": "string", "description": "The tail entity span should be a continuous span in the text provided in the input." } } } } } }
Output: { "relational triplets": [ { "head entity category": "People", "head entity span": "Rutherford B. Hayes", "relation": "Live in", "tail entity category": "Location", "tail entity span": "Delaware, Ohio" } ] }

- RE (without entity categories)

---

**Prompt:** [definition: {definition}\ntext: {text}\nrelations: {relations}\nrelational triplets:, {relational triplets}]

**TextTuning Example:**
**Input:** definition: Given a text and relations, you are required to scan the text and identify a list of relational triplets in it. Each relational triplet contains a head entity span, a relation, and a tail entity span. The head entity is the subject from which the relation originates. The relation represents the specific relation between the head entity and the tail entity. The tail entity is the object which the relation points. An entity span refers to the specific portion of the text that represents an entity.\ntext: The Peasants is a novel written by Nobel Prize-winning Polish author Wadysaw Reymont in four parts between 1904 and 1909.\nrelations: place served by transport hub, winner, field of work, location of formation, occupant\nrelational triplets:
**Output:** [["Nobel Prize", "winner", "Wadysaw Reymont"]]

**JsonTuning Example:**
**Input:** {"input": { "definition": "Given a text and relations, you are required to scan the text and identify a list of relational triplets in it. Each relational triplet contains a head entity span, a relation, and a tail entity span. The head entity is the subject from which the relation originates. The relation represents the specific relation between the head entity and the tail entity. The tail entity is the object which the relation points. An entity span refers to the specific portion of the text that represents an entity.", "text": "The Peasants is a novel written by Nobel Prize-winning Polish author Wadysaw Reymont in four parts between 1904 and 1909.", "relations": [ "place served by transport hub", "winner", "field of work", "location of formation", "occupant" ], "instruction": "{definition}\ntext: {text}\nrelations: {relations}\nrelational triplets: {relational triplets}" }, "output control": { "relational triplets": "type": "array", "items": { "type": "object", "properties": { "head entity span": { "type": "string", "description": "The head entity span should be a continuous span in the text provided in the input." }, "relation": { "type": "string", "description": "The relation should be one of the relations provided in the input." }, "tail entity span": { "type": "string", "description": "The tail entity span should be a continuous span in the text provided in the input." } } } } }
**Output:** { "relational triplets": [ { "head entity span": "Nobel Prize", "relation": "winner", "tail entity span": "Wadysaw Reymont" } ] }

---

- EE

Prompt: [definition: {definition}\ntext: {text}\nevent categories: {event categories}\nargument categories: {argument categories}\nevents:, {events}]

TextTuning Example:
Input: definition: Given a text, event categories, and argument categories, you are expected to scan the text and identify a list of events in it. Each event contains an event category, an event trigger, and a list of arguments. Each argument contains an argument category and an argument span. An event category represents the type of an event. An event trigger is the word or phrase in the text that explicitly denotes the occurrence of an event. Arguments are entities associated with an event and play specific roles or functions in relation to the event. An argument span refers to the specific portion of the text that represents an argument. An argument category refers to the category to which an argument belongs.\ntext: Until Basra, U.S. and British troops had encountered little resistance, even when they seized nearby Umm Qasr, and moved to secure key oil fields.\nevent categories: transfer money, start organization, extradite, meet, appeal, attack, convict, born, execute, transport, release parole, merge organization, sentence, divorce, end position, end organization, transfer ownership, start position, injure, sue, die, trial hearing, marry, nominate, charge indict, elect, declare bankruptcy, phone write, acquit, arrest jail, pardon, demonstrate, fine\nargument categories: instrument, vehicle, agent, seller, place, beneficiary, organization, destination, plaintiff, person, giver, recipient, victim, target, defendant, origin, prosecutor, entity, attacker, artifact, buyer, adjudicator\nevents:",
Output: [["attack", "seized", [["attacker", "troops"], ["place", "Umm Qasr"]]]]

JsonTuning Example:
Input: {"input": { "definition": "Given a text, event categories, and argument categories, you are expected to scan the text and identify a list of events in it. Each event contains an event category, an event trigger, and a list of arguments. Each argument contains an argument category and an argument span. An event category represents the type of an event. An event trigger is the word or phrase in the text that explicitly denotes the occurrence of an event. Arguments are entities associated with an event and play specific roles or functions in relation to the event. An argument span refers to the specific portion of the text that represents an argument. An argument category refers to the category to which an argument belongs.", "text": "Until Basra, U.S. and British troops had encountered little resistance, even when they seized nearby Umm Qasr, and moved to secure key oil fields.", "event categories": [ "transfer money", "start organization", "extradite", "meet", "appeal", "attack", "convict", "born", "execute", "transport", "release parole", "merge organization", "sentence", "divorce", "end position", "end organization", "transfer ownership", "start position", "injure", "sue", "die", "trial hearing", "marry", "nominate", "charge indict", "elect", "declare bankruptcy", "phone write", "acquit", "arrest jail", "pardon", "demonstrate", "fine" ], "argument categories": [ "instrument", "vehicle", "agent", "seller", "place", "beneficiary", "organization", "destination", "plaintiff", "person", "giver", "recipient", "victim", "target", "defendant", "origin", "prosecutor", "entity", "attacker", "artifact", "buyer", "adjudicator" ], "instruction": "definition: {definition}\ntext: {text}\nevent categories: {event categories}\nargument categories: {argument categories}\nevents: events}" }, "output control": { "events": { "type": "array", "items": { "type": "object", "properties": { "event category": { "type": "string", "description": "The event category should be one of the event categories provided in the input." }, "event trigger": { "type": "string", "description": "The event trigger should be a continuous span in the text provided in the input." }, "arguments": { "type": "array", "items": { "type": "object", "properties": { "argument category": { "type": "string", "description": "The argument category should be one of the argument categories provided in the input." }, "argument span": { "type": "string", "description": "The argument span should be a continuous span in the text provided in the input." } } } } } } } } }
Output: { "events": [ { "event category": "attack", "event trigger": "seized", "arguments": [ { "argument category": "attacker", "argument span": "troops" }, { "argument category": "place", "argument span": "Umm Qasr" } ] } ] }

- TQA

Prompt: [definition: {definition}\nquestion: {question}\ntable: {table}\nanswer:, {answer}]

TextTuning Example:
Input: definition: Given a question and a table, the task aims to output a list of values in the table to answer the question.\nquestion: which is the busiest domestic route out of houston intercontinental airport that does not have an american flight?\ntable: header : Rank — City — Passengers — Top Carriers row 1 : 1 — Los Angeles, CA — 700,000 — American, Spirit, United row 2 : 2 — Chicago, IL — 673,000 — American, Spirit, United row 3 : 3 — Denver, CO — 654,000 — Frontier, Spirit, United row 4 : 4 — San Francisco, CA — 492,000 — United row 5 : 5 — Dallas/Fort Worth, TX — 488,000 — American, United row 6 : 6 — Newark, NJ — 480,000 — United row 7 : 7 — Las Vegas, NV — 442,000 — Spirit, United row 8 : 8 — Charlotte, NC — 441,000 — United, US Airways row 9 : 9 — Atlanta, GA — 400,000 — Delta, United row 10 : 10 — Phoenix, AZ — 393,000 — United, US Airways\nanswer:
Output: Denver, CO

JsonTuning Example:
Input: {"input": { "definition": "Given a 'question' and a 'table', the task aims to output a list of values in the table to answer the question.", "question": "which is the busiest domestic route out of houston intercontinental airport that does not have an american flight?", "table": { "header": [ "Rank", "City", "Passengers", "Top Carriers" ], "rows": [ [ "1", "Los Angeles, CA", "700,000", "American, Spirit, United" ], [ "2", "Chicago, IL", "673,000", "American, Spirit, United" ], [ "3", "Denver, CO", "654,000", "Frontier, Spirit, United" ], [ "4", "San Francisco, CA", "492,000", "United" ], [ "5", "Dallas/Fort Worth, TX", "488,000", "American, United" ], [ "6", "Newark, NJ", "480,000", "United" ], [ "7", "Las Vegas, NV", "442,000", "Spirit, United" ], [ "8", "Charlotte, NC", "441,000", "United, US Airways" ], [ "9", "Atlanta, GA", "400,000", "Delta, United" ], [ "10", "Phoenix, AZ", "393,000", "United, US Airways" ] ] } }, "output control": { "answer": { "type": "string" } } }
Output: { "answer": "Denver, CO" }

• NL2SQL

> Prompt: [definition: {definition}\nquestion: {question}\ndatabase schema: {database schema}\nSQL query:, {SQL query}]
>
> TextTuning Example:
> Input: definition: Given a question and database schema that consists of table names and column names in the database, the text-to-SQL parsing task aims to translate the natural language question to a sql query that can be executed on the database to produce answers.\nquestion: List the title of all cartoons in alphabetical order.\ndatabase schema: Table: tv_channel; Columns: id, series_name, country, language, content, pixel_aspect_ratio_par, hight_definition_tv, pay_per_view_ppv, package_option. Table: tv_series; Columns: id, episode, air_date, rating, share, 18_49_rating_share, viewers_m, weekly_rank, channel. Table: cartoon; Columns: id, title, directed_by, written_by, original_air_date, production_code, channel\nSQL query:
> Output: select title from cartoon order by title
>
> JsonTuning Example:
> Input: {"input": { "definition": "Given a 'question' and 'database schema' that consists of table names and column names in the database, the text-to-SQL parsing task aims to translate the natural language question to a sql query that can be executed on the database to produce answers.", "question": "List the title of all cartoons in alphabetical order.", "database schema": [ { "table name": "tv_channel", "column names": [ "id", "series_name", "country", "language", "content", "pixel_aspect_ratio_par", "hight_definition_tv", "pay_per_view_ppv", "package_option" ] }, { "table name": "tv_series", "column names": [ "id", "episode", "air_date", "rating", "share", "18_49_rating_share", "viewers_m", "weekly_rank", "channel" ] }, { "table name": "cartoon", "column names": [ "id", "title", "directed_by", "written_by", "original_air_date", "production_code", "channel" ] } ] }, "output control": { "SQL query": { "type": "string" } } }
> Output: { "SQL query": "select title from cartoon order by title" }

## E.2 ROBUSTNESS

For evaluation, we employ 10 prompts for the MMLU benchmark, the NER task, and the NL2SQL task. Prompts are as follows:

• MMLU

> Prompt 1: [{question}\n{options_}\nAnswer:, {answer}]
> Prompt 2: [{question}\n\n{options_}\nAnswer:, {answer}]
> Prompt 3: [{question}\n{options_}, {answer}]
> Prompt 4: [Q: {question}\n\n{options_}\nA:, {answer}]
> Prompt 5: [Answer the following question: {question}\n\n{options_}\nAnswer:, {answer}]
> Prompt 6: [{options_}\n\n{question}\nAnswer:, {answer}]
> Prompt 7: [{options_}\nQ: {question}\nA:, {answer}]
> Prompt 8: [{question}\n\n{options_}\nThe answer is:, {answer}]
> Prompt 9: [{options_}\nGiven those answer options, answer the question: {question}\nAnswer:, {answer}]
> Prompt 10: [Q: {question}\n\n{options_}\nThe answer is:, {answer}]

• NER
  See Appendix D.

- NL2SQL

Prompt 1: [definition: {definition}\nquestion: {question}\ndatabase schema: {database schema}\nSQL query:, {SQL query}]
Prompt 2: [definition: {definition}\ndatabase schema: {database schema}\nquestion: {question}\nSQL query:, {SQL query}]
Prompt 3: [question: {question}\ndefinition: {definition}\ndatabase schema: {database schema}\nSQL query:, {SQL query}]
Prompt 4: [question: {question}\ndatabase schema: {database schema}\ndefinition: {definition}\nSQL query:, {SQL query}]
Prompt 5: [database schema: {database schema}\nquestion: {question}\ndefinition: {definition}\nSQL query:, {SQL query}]
Prompt 6: [database schema: {database schema}\ndefinition: {definition}\nquestion: {question}\nSQL query:, {SQL query}]
Prompt 7: [{definition}\nquestion: {question}\ndatabase schema: {database schema}\nSQL query:, {SQL query}]
Prompt 8: [{definition}\ndatabase schema: {database schema}\nquestion: {question}\nSQL query:, {SQL query}]
Prompt 9: [question: {question}\ndatabase schema: {database schema}\n{definition}\nSQL query:, {SQL query}]
Prompt 10: [database schema: {database schema}\nquestion: {question}\n{definition}\nSQL query:, {SQL query}]

