# OpenReview forum: "JsonTuning: Towards Generalizable, Robust, and Controllable Instruction Tuning"
_ICLR.cc/2024/Conference — Submitted to ICLR 2024_

### Official Review · Reviewer_5Thw · 2023-10-29

**Soundness:** 2 fair
**Presentation:** 3 good
**Contribution:** 2 fair
**Rating:** 5
**Confidence:** 4

**Summary:**

The authors propose JsonTuning which unifies different text-to-text tasks into a structured format, i.e. json format. Experiments and analyses are conducted on various popular benchmarks with models from the LLaMA family ranging from 7B to 13B.

**Strengths:**

* The motivation is simple and easy to understand.
* The paper is well-written and easy to follow.
* Comprehensive experiments and analyses are provided to support the authors' claims.

**Weaknesses:**

There are several key limitations of the proposed approach:
* Context window consumption: JsonTuning demands much more tokens than TextTuning and it poses a challenge both in the training and inference time. One key aspect of current LLMs is their context window, and how to use it efficiently. The authors should provide evidence relating to the extra overhead on training/inference of JsonTuning.
*  It is not also obvious to transform the user's request into a structured format. Despite the authors' arguments on the generalization capability of JsonTuning, structured object like Json still lags far behind natural language in terms of expressivity. The tasks concerned in the work are relatively easy and can be transformed into json format with less effort. However, I am not convinced this generalizes to real industrial use cases where the user's request could be far more complex.
* A simple question: assume a user requests the model to produce output following json format at the first place. Then how the output control be defined on the input side? I suppose it would be a nested json and could be fragile for the model to take as input.

**Questions:**

Please see the weakness section.

---

> ### Author Response · Authors · 2023-11-20
> **Response to Reviewer 5Thw**
>
> > Analysis of the tokens used for JsonTuning and TextTuning
>
> We have calculated the input and output tokens used for each training and evaluation example in both TextTuning and JsonTuning. The average tokens of all training and evaluation examples for these methods are presented in the following table:
>
> |  | Training | Evaluation |
> | -------- | -------- | -------- |
> | Text w/ label space and control information | 317.53 | 236.24 |
> | Json w/ label space and control information | 394.61 | 296.55 |
> | Text w/o label space and control information | 278.17 | 183.91 |
> | Json w/o label space and control information | 349.23 | 235.16 |
>
> The table shows that JsonTuning uses approximately 25% more tokens for training and evaluation compared to TextTuning. The additional tokens are primarily used to encode the JSON structured format.
>
> There are potential ways to reduce the tokens used for JsonTuning:
>
> * JsonTuned models are robust to different prompts, so using lengthy prompts may not be necessary. We can replace them with concise prompts.
> * JsonTuning offers the flexibility to adjust the structure. Simplifying the input and output structures could be helpful, as JsonTuning requires more tokens for more complex structures.
>
> > Regarding the expressivity of natural language texts and JSON structures
>
> The expressivity of the constructed JSON structures is at least equivalent to that of their corresponding natural language texts, as JSON structures can easily be converted into natural language texts. For instance, consider the following JSON input and text input. By replacing placeholders, such as {question} and {options}, in the JSON structures with their respective values, the text input can be obtained.
>
> **JSON Input**: {"input": {"question": "Who is the CEO of Google?", "options": "(A) Sundar Pichai (B) Bill Gates (C) Tim Cook (D) Satya Nadella", "instruction": "Answering the following question: {question} {options}. Answer: {answer}"}}
>
> **Text Input**: "Answering the following question: Who is the CEO of Google? (A) Sundar Pichai (B) Bill Gates (C) Tim Cook (D) Satya Nadella. Answer:"
>
> However, this transformation may lose some information, such as what are the key elements of the task. Our proposed JsonTuning method aims to address this issue. Consequently, JSON structures offer more valuable information than natural language texts, resulting in higher expressivity.
>
> > Regarding transforming the user's request into a structured format, especially for complex use cases
>
> Transforming user requests into a structured JSON format can be efficiently achieved by developing a user interface that enables users to input key task elements, task prompts, and control information. This approach ensures that users can effectively communicate their requirements without needing to understand JSON structures. As the input and output JSON structures are consistent across all tasks, we can automatically construct them based on the user inputs. For more complex use cases, the user interface can guide the model in identifying the essential information for the target task and the desired outputs.
>
> JsonTuning offers several advantages for applications:
>
> * Robustness: The method's resilience to varying prompts significantly reduces the need for users to test multiple prompts for optimal performance. If users do not wish to provide a task prompt, a default prompt can be generated based on the key task elements they supply, thereby minimizing user effort. Furthermore, as backend models are updated or changed, the optimal prompt for natural language texts may vary, necessitating adjustments for each model version. In contrast, Json-tuned models may not require prompt modifications even when the backbone model is updated, considerably reducing user effort.
>
> * Output Control: The output control component allows users to accurately specify and easily parse the output, which can be challenging when using natural language texts. Users only need to comprehend basic types such as string, object, and array. The descriptions and examples of these data types can be provided through the user interface. This approach necessitates minimal human effort, and the underlying JSON structures can be automatically constructed once users indicate the output elements they require.
>
> By implementing JsonTuning in real-world applications, a more professional and efficient user experience can be achieved.

---

> > ### Author Response · Authors · 2023-11-20
> > **Response to Reviewer 5Thw (Part 2)**
> >
> > > Assume a user requests the model to produce output following json format at the first place. Then how the output control be defined on the input side? I suppose it would be a nested json and could be fragile for the model to take as input.
> >
> > Output control in JsonTuning is specifically designed to accommodate users' unique requirements, enabling a more accurate representation of their intentions compared to TextTuning. Consequently, defining the output control relies on the users' specific objectives. For instance, in a medical diagnosis task that aims to identify diseases based on certain symptoms, the output could be an array of potential diseases, the most probable disease, or diseases associated with particular organs such as the heart or lungs.
> >
> > Regarding your concern about the complexity of the output control, it is crucial to acknowledge that the complexity of the output control depends on the particular use case and user requirements. A well-trained LLM should effectively manage various output structures with diverse complexities. To achieve this, It is essential to train the LLM on diverse examples, including various JSON structures and output control patterns, to ensure its adaptability to different patterns and generate the desired output. This is why we introduce InstrucUIE for training, as it allows the model to handle complex structures effectively. Our evaluation results on the EE task demonstrate that the Json-tuned model can generalize to more complex structures even when trained only on simpler structures. Furthermore, our case studies on controllability illustrate that the model can generate the output in compliance with the output control effectively.

---

### Official Review · Reviewer_qRB3 · 2023-11-01

**Soundness:** 3 good
**Presentation:** 3 good
**Contribution:** 3 good
**Rating:** 5
**Confidence:** 4

**Summary:**

The paper introduces a new approach called JsonTuning, which leverages the structured JSON format to represent tasks. JsonTuning improves generalization by helping the model understand task elements and their relationships, enhances robustness by reducing ambiguity, and offers greater control over the model's output. The paper presents a thorough comparative study with different language models and evaluation benchmarks, demonstrating that JsonTuning surpasses TextTuning in various applications, achieving better performance, adaptability, robustness, and control.

**Strengths:**

This paper provides a very simple method to convert the original instruction tuning into a unified Json format. The authors also conduct comprehensive comparison against baselines trained under text-to-text formulation, and show that JsonTuning can harvest better performance. The ablation studies help us to better understand the effect of the subparts such as label space and control information.

**Weaknesses:**

Despite the commendable performance exhibited by JsonTuning, there are still several notable weaknesses:

**Introduction of Additional Knowledge in Input**: The utilization of Json formatting can aid models in generating outputs that conform to the constraints specified in the Json input. However, it also introduces supplementary knowledge, such as information pertaining to input and output types, which is originally absent in the unstructured textual instructions. To facilitate a fair comparison, it becomes essential to incorporate the information present in the Json input but absent in the textual input into the original text instructions. This allows for a comprehensive assessment of whether the Json format indeed outperforms the text format.

**Potential Incompleteness of Generalization Experiments**: While the training dataset encompasses a diverse range of tasks, including Flan and P3++, there appears to be some overlap between the types of tasks assessed during testing and those encountered in training. It is pertinent to explore whether the model can adeptly generalize to entirely unseen task types, which may not have been covered by the training data. Furthermore, the evaluation tasks, primarily centered on structure prediction, demand output format constraints. However, for the field of instruction tuning, there exists a research interest in user-oriented instruction following evaluation, exemplified by AlpacaEval. In cases involving more open-ended instruction data, the Json format may not bring additional benefits since there are no specific constraints imposed on the output format. It also remains uncertain how the model performs on the Super-NI test set, a widely employed dataset assessing a model's ability to follow instructions.

**Challenges for Real-World Applications**: The practical applicability of Json-tuned models is constrained by the need to manually craft complex Json prompts that delineate output fields and their corresponding types. For users lacking expertise in computer science, utilizing such models proves challenging, as they may lack the proficiency required to construct a Json-formatted instruction. Unless a method for automatically generating Json instructions becomes available, the utility of this approach is notably limited in real-world scenarios.

**Questions:**

1. Will there be a significant improvement after fine-tuning code LLMs (e.g., CodeLLAMA) on the Json instruction data? It seems that code LLMs might be a better counterpart that learns the Json data more easily.

2. There might be several missing relevant references. The first one have studied Json-format instruction data. The second one is a resource that has the similar formulation with Super-NI and can be represented in well-structured Json format as well:
- Did You Read the Instructions? Rethinking the Effectiveness of Task Definitions in Instruction Learning. Yin et al., 2023.
- Dynosaur: A Dynamic Growth Paradigm for Instruction-Tuning Data Curation. Yin et al., 2023.

---

> ### Author Response · Authors · 2023-11-20
> **Responses to Reviewer qRB3**
>
> > Introduction of Additional Knowledge in Input
>
> In Section 4: Analysis of our paper, we conduct an ablation study of JsonTuning without label space and control information, where JsonTuning still outperforms TextTuning in terms of generalization. To address your question comprehensively, we perform additional experiments with TextTuning, incorporating label space and control information. The generalization and robustness results on LLaMA-7b are as follows:
>
> Generalization Results:
>
> | Model | MMLU | BBH | NER | RE | EE | TQA | NL2SQL | Average |
> | -------- | -------- | -------- | -------- | -------- | -------- | -------- | -------- | -------- |
> | Text w/ label space and control information | 44.56 | 36.77 | 35.70 | 15.77 | 1.45 / 0.00 | 16.40 | 14.60 | 23.50 |
> | Json w/ label space and control information | 44.69 | 37.08 | 41.80 | 15.56  | 3.09 / 8.24 | 16.40 | 16.40 | **25.37**
> | Text w/o label space and control information | 43.11 | 32.48 | 36.47 | 13.60 | 1.08 / 0.00 | 18.20 | 8.60 | 21.86
> | Json w/o label space and control information | 42.66 | 36.23 | 47.65 | 16.23 | 0.99 / 0.29 | 10.40 | 12.80 | **23.80**
>
> Robustness results on MMLU with 10 different prompts (we calculate the mean and standard deviation (Std) of the performance using these 10 prompts.):
>
> | Model | Mean | Std |
> | -------- | -------- | -------- |
> | Text w/ label space and control information | 42.02 | 3.31 |
> | Json w/ label space and control information | **44.63** | **0.11** |
> | Text w/o label space and control information | 38.82 | 6.28 |
> | Json w/o label space and control information | **42.19** | **0.21** |
>
> We can make the following observations: (1) JsonTuning consistently surpasses TextTuning in both generalization and robustness, regardless of the inclusion of label space and control information. Importantly, JsonTuning significantly reduces the model's sensitivity to different prompts, as evidenced by the lower variance in performance across prompts. (2) TextTuning with control information still struggles to generalize well to more complex structured tasks, as illustrated by the poor performance on the EE task. JsonTuning demonstrates a marked improvement in this regard. These findings emphasize the importance of structured formats in instruction tuning. Furthermore, our proposed components – label space and control information – consistently enhance the model's generalization capabilities and robustness, highlighting their value in instruction tuning.
>
> > Potential Incompleteness of Generalization Experiments
>
> Our evaluation encompasses various unseen tasks and datasets. While some task types, such as multiple-choice QA, are encountered during training, the specific evaluation tasks and datasets remain unseen. Furthermore, tasks like EE and TQA represent entirely novel task types. Our JsonTuning method allows models to generalize to more complex EE structures, even if they were not encountered during training, demonstrating the robust generalization capabilities of our Json-tuned models.
>
> We conduct an evaluation of LLaMA-7B-Json and LLaMA-7B-Text on AlpacaEval. Both models were trained with label space and control information. To assess LLaMA-7B-Json, we construct an input JSON structure for a natural language question Q, represented as {"input": {"question": Q, "instruction": "{question}\n{answer}"}, "output control": {"answer": {"type": "string"}}}. The win rates for LLaMA-7B-Json and LLaMA-7B-Text are 4.78% and 4.64%, respectively. The slightly better performance of LLaMA-7B-Json suggests that JsonTuning may offer additional advantages in open-ended scenarios.
>
> Since the Super-NI test set is already incorporated into the Flan2022 collection, it would not be appropriate to evaluate models on this test set, as many instances in it are seen during training.

---

> > ### Author Response · Authors · 2023-11-20
> > **Responses to Reviewer qRB3 (Part 2)**
> >
> > > Challenges for Real-World Applications
> >
> > Our JsonTuning method can be readily applied to real-world scenarios. Transforming user requests into a structured JSON format can be efficiently achieved by developing a user interface that enables users to input key task elements, task prompts, and control information. This approach ensures that users can effectively communicate their requirements without needing to understand JSON structures. As the input and output JSON structures are consistent across all tasks, we can automatically construct them based on the user inputs.
> > JsonTuning offers several advantages for applications:
> >
> > * Robustness: The method's resilience to varying prompts significantly reduces the need for users to test multiple prompts for optimal performance. If users do not wish to provide a task prompt, a default prompt can be generated based on the key task elements they supply, thereby minimizing user effort. Furthermore, as backend models are updated or changed, the optimal prompt for natural language texts may vary, necessitating adjustments for each model version. In contrast, Json-tuned models may not require prompt modifications even when the backbone model is updated, considerably reducing user effort.
> >
> > *  Output Control: The output control component allows users to accurately specify and easily parse the output, which can be challenging when using natural language texts. Users only need to comprehend basic types such as string, object, and array. The descriptions and examples of these data types can be provided through the user interface. This approach necessitates minimal human effort, and the underlying JSON structures can be automatically constructed once users indicate the output elements they require.
> >
> > By implementing JsonTuning in real-world applications, a more professional and efficient user experience can be achieved.
> >
> > > Will there be a significant improvement after fine-tuning code LLMs (e.g., CodeLLAMA) on the Json instruction data? It seems that code LLMs might be a better counterpart that learns the Json data more easily.
> >
> > Code LLMs may help learn the JSON format but may not significantly improve the performance in general as the performance heavily depends on the base model. While code LLMs excel in code-related tasks, their impact on other tasks may vary. The results of LLaMA2-7B and CodeLLaMA-7B, which is trained based on LLaMA2-7B, are as follows:
> >
> > | Model | MMLU | BBH | NER | RE | EE | TQA | NL2SQL | Average |
> > | -------- | -------- | -------- | -------- | -------- | -------- | -------- | -------- | -------- |
> > | LLaMA2-7B | 47.95 | 39.23 | 43.68 | 24.71 | 4.00 / 4.16 | 20.00 | 11.20 | 27.26 |
> > | CodeLLaMA-7B | 41.34 | 40.49 | 47.52 | 22.12 | 4.75 / 9.95 | 15.80 | 41.60 | 30.89 |
> >
> > As anticipated, CodeLLaMA-7B exhibited a substantial improvement over LLaMA2-7B in the NL2SQL task, with performance increasing from 11.20 to 41.60. However, it performed worse on MMLU, indicating continual pre-training on code may hurt the model's performance on general tasks.
> >
> > > Regarding the missing references
> >
> > Thank you for your suggestion. The first paper, titled "Did You Read the Instructions? Rethinking the Effectiveness of Task Definitions in Instruction Learning," focuses on understanding the importance of different parts of a task definition and identifying key elements in task definitions. The second paper, "Dynosaur: A Dynamic Growth Paradigm for Instruction-Tuning Data Curation," proposes a paradigm for automatically curating instruction-tuning data, which can be represented in a well-structured JSON format.
> >
> > We acknowledge the valuable contributions of both papers in the instruction tuning field and their relevance to our work. We will cite these papers in the updated version of our paper.

---

> > > ### Comment · Reviewer_qRB3 · 2023-11-21
> > >
> > > I greatly appreciate the author's comprehensive response to my review. However, I still have some concern about 1) the practical use, 2) in-depth analysis, and 3) novelty of the research.
> > >
> > > In terms of the practical use, though you mention that it is possible to design a user interface and then convert the user input into instruction-tuning data format. However, different task types may involve distinct Json data fields. Once there comes a new task type that accepts completely different information, the user interface will have to be updated manually. Also, it is not realistic to require users to provide very detailed instruction and examples specifically for JsonTuning data, as people always prefer fully automatic and general approaches that guarantee the data quality.
> > >
> > > Regarding the new experimental results, it is shown that on some tasks such as TQA, "Json w/o label space and control information" underperforms the TextTuning counterparts by a large margin. Though the average performance is good, it is still unclear to me why JsonTuning has significantly worse performance than TextTuning on this task type. Also, if we remove the tasks such as NER and NL2SQL, the average performance may become lower than the text-only counterparts. More in-depth analysis (e.g., the error case study) is needed.
> > >
> > > For the novelty, the Section 5 in Yin et al., 2023 ("Did You Read the Instructions? Rethinking the Effectiveness of Task Definitions in Instruction Learning") shows the benefits of introducing Json formats in instruction tuning. The paper discusses something beyond identifying the key elements in task definitions. The authors might put more emphasis on how to highlight the distinct contributions not covered in the prior works.

---

> > > > ### Author Response · Authors · 2023-11-22
> > > > **Response to Reviewer qRB3**
> > > >
> > > > > Regarding the practical use
> > > >
> > > > We would like to clarify that the user interface for JsonTuning can be designed to accommodate various task types without the need for manual updates, as all tasks have unified input and output structures in JsonTuning. Users are allowed to input the key elements of the task, and there is no need to define these task fields manually. Furthermore, users do not need to provide highly detailed instructions and examples specifically for JsonTuning. In contrast, Json-tuned models are robust to different prompts, eliminating the need for users to provide a well-designed prompt, which is often required when using natural language texts for solving tasks. If users do not wish to provide a task prompt, a default prompt can be generated based on the key task elements they supply, thereby minimizing user effort.
> > > >
> > > > For example, if users want to tackle a multiple-choice QA task, traditionally, they might provide the following text input:
> > > >
> > > > *Answer the following question: Who is the CEO of Google? (A) Sundar Pichai (B) Bill Gates (C) Tim Cook (D) Satya Nadella. Answer:*
> > > >
> > > > For using Json-tuned models, they may only need to provide the following key task elements (including their names and contents):
> > > >
> > > > *question: Who is the CEO of Google?*
> > > >
> > > > *options: (A) Sundar Pichai (B) Bill Gates (C) Tim Cook (D) Satya Nadella*
> > > >
> > > > *answer: (the desired output, no need to provide the content)*
> > > >
> > > > The user interface guides users to identify the key elements required for solving the task clearly. If users prefer to provide a prompt, they may input *"Answer the following question: {question} {options}. Answer: {answer}."* If not, a default prompt can be constructed based on the names of the key task elements provided by the users. This default prompt might be *"{question} {options} \n {answer}."* Since JsonTuned models are robust to different prompts, using the default prompt may not significantly impact performance. For more complex tasks, users often need to refine their prompts manually multiple times to achieve satisfactory results. JsonTuning considerably reduces users' efforts in this regard.
> > > >
> > > > After users provide the key task elements and the prompt, the corresponding JSON structure can be automatically generated. In an extreme case, if users do not want to provide key elements but want to use Json-tuned models like text-tuned models, this is also achievable, as the entire natural language text can be considered a key element of the task. A default name for this element, such as "input" or "question," can be assigned, and a default input JSON structure can be created to solve the task.
> > > >
> > > > In conclusion, the effort required for using Json-tuned models is equivalent to or lower than using natural language texts to express users' requests, making it a more efficient and user-friendly approach.
> > > >
> > > > > Regarding In-depth Analysis
> > > >
> > > > The TQA task is defined as follows: "Given a question and a table, the task aims to output a list of values in the table to answer the question." In the absence of control information, the model is more likely to generate invalid JSON structures and produce inconsistent output types. Specifically, 18.2% of the outputs consist of invalid JSON structures. Among the valid JSON structures, the answer types comprise strings (34.0%), arrays (62.6%), and arrays of arrays (3.4%), which presents evaluation challenges and negatively impacts performance. By providing control information, all outputs are valid JSON structures. Therefore, control information is essential for enhancing the clarity and accuracy of the generated responses.
> > > >
> > > > > Regarding the novelty of our work
> > > >
> > > > The novelty and contribution of our work lie in the application of the structure-to-structure paradigm to instruction tuning, which enhances the generalization ability, robustness, and controllability of instruction-tuned models. Our approach focuses on identifying key task elements, distinguishing between task elements and instructions, and reducing ambiguity.
> > > >
> > > > The paper titled "Did You Read the Instructions? Rethinking the Effectiveness of Task Definitions in Instruction Learning" demonstrates the benefits of structuring task definitions within the text-to-text paradigm. Both this paper and our work emphasize the importance of structure in instruction tuning. In our research, task definition is considered a task element. The aforementioned paper suggests that further structuring of task definitions may lead to improved performance, thereby highlighting the potential of incorporating more fine-grained structures in instruction tuning. We appreciate your suggestion and will discuss this in the related work section of our paper.
> > > >
> > > >
> > > >
> > > > **We would greatly appreciate it if you could consider increasing your rating**. We are committed to continuously improving our work and engaging in further discussions. Should you have any additional concerns or suggestions, we will do our utmost to address them in a timely and thorough manner.

---

> > > > > ### Comment · Reviewer_qRB3 · 2023-12-04
> > > > >
> > > > > Thanks for the comprehensive response! However, in terms of practical use, I still think that it is not realistic to ask a user, especially a non-expert user to volunteer to type information. Also, the response still doesn't address my concern if the JsonTuning model is applied to other instruction-tuning benchmarks such as AlpacaEval/VicunaEval. I may still keep my score.

---

### Official Review · Reviewer_jpTc · 2023-11-01

**Soundness:** 3 good
**Presentation:** 3 good
**Contribution:** 2 fair
**Rating:** 6
**Confidence:** 5

**Summary:**

This paper uses JSON as a formatting scheme to tune language models, and shows that this can help models learn and generalize better than formatting examples as plain text. JSONTuning also reduces prompt sensitivity.

**Strengths:**

- The paper is clear and easy to follow.
 - The method is simple and works well.

**Weaknesses:**

- In my opinion this paper needs some sort of analysis of the number of additional tokens introduced by the JSON format for training and inference. The additional training cost is probably negligible and unimportant, but the additional FLOPS and encoded/decoded tokens for the JSON format will add up for inference. Note: I am not saying that the fact that JSON-formatted examples have extra tokens is a weakness, but this extra cost should at least be quantified in my opinion.
 - I generally don't think papers should be penalized harshly for a lack of novelty or for overly simple methods, but it must be acknowledged that the idea of "format exampels as JSON instead of plain text" can only be taken so far. Note: I am not saying this is grounds for rejection, just that this paper's impact and contribution is limited.

**Questions:**

- How do text-tuned models behave as a base for JSON-formatted examples? For example, I think some of the following questions should be answered or at least discussed:
   * What if we text-tune the model and then JSONTune it on a small dataset?
   * What if we text-tune the model and then use JSON-formatted examples in a few shot prompt?
 - What is the decoding mechanism used for evaluating models? I assume it is just greedy sampling, but it would be nice for more evaluation details to be in the appnendix.
 - How often was invalid JSON generated? What do the authors think about constrained decoding schemes like [Scholak et al](https://arxiv.org/abs/2109.05093), is there any point of using constrained decoding to improve JSONTuned models?

---

> ### Author Response · Authors · 2023-11-20
> **Response to Reviewer jpTc**
>
> > Analysis of the tokens used for JsonTuning and TextTuning
>
> We have calculated the input and output tokens used for each training and evaluation example in both TextTuning and JsonTuning. The average tokens of all training and evaluation examples for these methods are presented in the following table:
>
> |  | Training | Evaluation |
> | -------- | -------- | -------- |
> | Text w/ label space and control information | 317.53 | 236.24 |
> | Json w/ label space and control information | 394.61 | 296.55 |
> | Text w/o label space and control information | 278.17 | 183.91 |
> | Json w/o label space and control information | 349.23 | 235.16 |
>
> The table shows that JsonTuning uses approximately 25% more tokens for training and evaluation compared to TextTuning. The additional tokens are primarily used to encode the JSON structured format.
>
> There are potential ways to reduce the tokens used for JsonTuning:
>
> * JsonTuned models are robust to different prompts, so using lengthy prompts may not be necessary. We can replace them with concise prompts.
> * JsonTuning offers the flexibility to adjust the structure. Simplifying the input and output structures could be helpful, as JsonTuning requires more tokens for more complex structures.
>
> > What if we text-tune the model and then Json-tune it on a small dataset?
>
> To explore this, we continue instruction-tuning the text-tuned model with JsonTuning using varying numbers of examples (120, 600, 1200, 6000, and 12000) while maintaining the relative data ratio of the Flan 2022 collection and structured tasks in our primary experiments. The LLaMA-7b results on MMLU and NL2SQL are as follows:
>
> |  | 120 | 600 | 1200 | 6000 | 12000 |
> | -------- | -------- | -------- |  -------- | -------- | -------- |
> | MMLU | 0.00 | 40.83 | 39.83 | 42.45 | 41.85 |
> | NL2SQL | 0.00 | 4.40 | 3.80 | 5.20 | 3.40 |
>
> The results of the original text-tuned model on MMLU and NL2SQL are 43.11 and 8.60, respectively.
>
> We observe that: (1) A very limited number of JSON-formatted examples (e.g., 120) is insufficient for the text-tuned model to handle JSON-formatted tasks. (2) Tuning the text-tuned model with a small set of JSON-formatted examples results in degraded performance. These observations suggest that using the text-tuned model as a base for JsonTuning is not appropriate.
>
> > What if we text-tune the model and then use JSON-formatted examples in a few shot prompt?
>
> This approach is ineffective. We sampled 100 MMLU examples and used 3 JSON-formatted examples as few-shot prompts. The text-tuned LLaMA-7b model still generates natural language texts instead of JSON structures and these texts seem non-sense. While a significantly larger text-tuned model may generate valid JSON structures, its performance could be substantially lower than the JSON-tuned model.
>
> > What is the decoding mechanism used for evaluating models? I assume it is just greedy sampling, but it would be nice for more evaluation details to be in the appendix.
>
> You are correct. We use greedy decoding. We appreciate your suggestion and will include more evaluation details in the updated paper.
>
> > How often was invalid JSON generated? What do the authors think about constrained decoding schemes like Scholak et al, is there any point of using constrained decoding to improve JSONTuned models?
>
> The proportion of invalid JSON structures generated by LLaMA-7B on evaluation tasks is as follows:
>
> | MMLU | BBH | NER | RE | EE | TQA | NL2SQL |
> | -------- | -------- | -------- | -------- | -------- | -------- | -------- |
> 0.00% | 1.57% | 5.74% | 0.27% | 4.19% | 0.00% | 3.60%
>
> As anticipated, tasks with intricate output structures, such as NER and EE, exhibit a higher ratio of invalid JSON structures compared to tasks with simpler output structures, such as MMLU and BBH.
>
> PICARD, a constrained decoding method proposed by Scholak et al., is designed to produce valid SQL queries by rejecting inadmissible tokens at each decoding step. In the context of our JsonTuning paradigm, employing a constrained decoding scheme that ensures the generation of valid output structures adhering to the output control could be advantageous. However, implementing such a method may be challenging, and it could potentially increase decoding time. As an alternative, we may utilize post-processing techniques to obtain valid JSON structures, as they are typically efficient and cost-effective. Previously, we treated invalid output structures as NULL, which might have been overly strict for evaluation purposes. Moving forward, we plan to investigate suitable post-processing techniques that can be broadly applied to various output structures, aiming to enhance the performance and validity of Json-tuned models.

---

### Official Review · Reviewer_pk2e · 2023-11-01

**Soundness:** 3 good
**Presentation:** 4 excellent
**Contribution:** 2 fair
**Rating:** 6
**Confidence:** 3

**Summary:**

This paper proposes JsonTuning,  a novel structure-to-structure approach for instruction tuning. It leverages the widely used JSON format and uses the structured format to define tasks. Empirical results show JsonTuning improves TextTuning on generalization, robustness, and controllability. Ablation study shows the label space and control information are both curial for JsonTuning. The paper also analyzes the effects of data size and structured tasks.

**Strengths:**

- Originality: The paper proposes JsonTuning, which is a new format of instruction tuning. It reformats standard instruction tuning data into JSON format to reduce ambiguity and improve controllability. The proposed method is novel in that it leverages the LLM's understanding of structured data format - JSON and makes use of it in downstream instruction tuning.
- Clarity: The paper is well-written and easy to follow.
- Quality: The claims in this paper are well supported by citations or empirical results. The authors clearly demonstrate JsonTuning's advantages in generalization, robustness, and controllability.

**Weaknesses:**

- Significance: the format of instruction tuning data is only one of the many system choices of the overall instruction tuning. Others include tasks, base model, domains, and languages. Since the paper only focuses on the data format on a selection of tasks, the significance is limited.
- Soundness: It is not clear if a TextTuning model with candidate answers and output control in plain text would also perform as good as JsonTuning. In other words, how much gain was from the structured format of JSON itself?

**Questions:**

- How did you construct training data for JsonTuning from plain text instruction tunning data? Did you use human annotator to do the conversion or use another language model?
- Why did you choose JSON format instead of other popular structured formats, such as XML and YAML?

---

> ### Author Response · Authors · 2023-11-20
> **Response to Reviewer pk2e**
>
> > Regarding the significance of our work
>
> The performance of instruction tuning is influenced by several key factors, including the base model, tuning data, and data format. While the base model and tuning data have received considerable attention from researchers, the significance of data format has often been overlooked. In this paper, we aim to emphasize the importance of data format in instruction tuning, focusing on its role in generalization, robustness, and output control.
>
> Our experiments and analysis demonstrate that a structured data format, such as JSON, is crucial for effective instruction tuning. First, our method, JsonTuning, consistently outperforms TextTuning across various base models and tuning data quantities, highlighting its generalizability and complementarity with other factors. Second, JsonTuning substantially enhances the robustness of instruction-tuned models, an improvement that is difficult to achieve solely through the base model and tuning data optimization. For instance, our robustness results indicate that upgrading the base model from LLaMA to LLaMA2 does not significantly enhance robustness; however, JsonTuning considerably improves it. Finally, JSON structures offer well-defined control information for output generation, a feature that is challenging to achieve using natural language text alone.
>
> > It is not clear if a TextTuning model with candidate answers and output control in plain text would also perform as well as JsonTuning. In other words, how much gain was from the structured format of JSON itself?
>
> In Section 4: Analysis of our paper, we conduct an ablation study of JsonTuning without label space and control information, where JsonTuning still outperforms TextTuning in terms of generalization. To address your question comprehensively, we perform additional experiments with TextTuning, incorporating label space and control information. The generalization and robustness results on LLaMA-7b are as follows:
>
> Generalization Results:
>
> | Model | MMLU | BBH | NER | RE | EE | TQA | NL2SQL | Average |
> | -------- | -------- | -------- | -------- | -------- | -------- | -------- | -------- | -------- |
> | Text w/ label space and control information | 44.56 | 36.77 | 35.70 | 15.77 | 1.45 / 0.00 | 16.40 | 14.60 | 23.50 |
> | Json w/ label space and control information | 44.69 | 37.08 | 41.80 | 15.56  | 3.09 / 8.24 | 16.40 | 16.40 | **25.37**
> | Text w/o label space and control information | 43.11 | 32.48 | 36.47 | 13.60 | 1.08 / 0.00 | 18.20 | 8.60 | 21.86
> | Json w/o label space and control information | 42.66 | 36.23 | 47.65 | 16.23 | 0.99 / 0.29 | 10.40 | 12.80 | **23.80**
>
> Robustness results on MMLU with 10 different prompts (we calculate the mean and standard deviation (Std) of the performance using these 10 prompts.):
>
> | Model | Mean | Std |
> | -------- | -------- | -------- |
> | Text w/ label space and control information | 42.02 | 3.31 |
> | Json w/ label space and control information | **44.63** | **0.11** |
> | Text w/o label space and control information | 38.82 | 6.28 |
> | Json w/o label space and control information | **42.19** | **0.21** |
>
> We can make the following observations: (1) JsonTuning consistently surpasses TextTuning in both generalization and robustness, regardless of the inclusion of label space and control information. Importantly, JsonTuning significantly reduces the model's sensitivity to different prompts, as evidenced by the lower variance in performance across prompts. (2) TextTuning with control information still struggles to generalize well to more complex structured tasks, as illustrated by the poor performance on the EE task. JsonTuning demonstrates a marked improvement in this regard. These findings emphasize the importance of structured formats in instruction tuning. Furthermore, our proposed components – label space and control information – consistently enhance the model's generalization capabilities and robustness, highlighting their value in instruction tuning.

---

> > ### Author Response · Authors · 2023-11-20
> > **Response to Reviewer pk2e (Part 2)**
> >
> > > How did you construct training data for JsonTuning from plain text instruction tunning data? Did you use human annotator to do the conversion or use another language model?
> >
> > We generate the training data automatically without relying on human annotators or language models for construction. Specifically, we perform the following tasks: (1) We modify the data generation code for the Flan2022 collection (https://github.com/google-research/FLAN/tree/main/flan/v2) to include the task input and output elements, prompt, and the label space if available and for each example. (2) For tasks with limited label space in subcollections of Flan2022 such as CoT and Super Natural Instructions v2, we construct the label space by gathering their possible task outputs, as their task sources do not provide explicit label spaces. (3) For the Flan2022 collection, we create JSON input and output structures based on the prompts used for each example, as they clearly indicate input and output elements, with all outputs being string types. For InstructUIE, we manually construct 10 prompts for each task and design the control information to build the input and output structures. We will make our processed data and data processing code publicly available.
> >
> > > Why did you choose JSON format instead of other popular structured formats, such as XML and YAML?
> >
> > Our proposed structure-to-structure paradigm can be implemented using various structured formats, including JSON, XML, and YAML. We selected JSON because (1) it has native support for various data types such as string, array, and object;  (2) its syntax is simpler and consistent, facilitating easy parsing and generation; (3) it is natively supported in most programming languages and is highly compatible with Python.
> > Nevertheless, the choice ultimately depends on specific use cases, requirements, and preferences. XML may be preferable for support of metadata, namespaces, or more complex data such as charts or images, while YAML could be suitable for prioritizing human readability and simplicity.

---

> > > ### Comment · Reviewer_pk2e · 2023-12-04
> > >
> > > Thank you for the detailed reply and updated results. I have read other reviews and the replies. I found certain value in the paper, but the significance and the generalization of the method make it a "weak" accept. I will keep my score unchanged.

---

### Author Response · Authors · 2023-11-20
**General Response**

Dear Reviewers,


We sincerely appreciate your insightful and constructive feedback on our submission. We have made every effort to address your concerns and respond to your questions thoroughly. Should you have any further inquiries or require additional clarification, we would be more than happy to engage in further discussion.

Thank you again for your valuable suggestions, and we look forward to your continued guidance.


Warm regards,

Authors of Submission 7027

---

### Author Response · Authors · 2023-11-22
**We would like to know if you have any further questions or require additional clarification.**

Dear Reviewers,

We appreciate the time and effort you have dedicated to reviewing our work and providing insightful feedback. We have thoroughly analyzed your comments and have formulated our responses accordingly.

Should you have any further inquiries or require additional clarification, please do not hesitate to inform us.
We would like to request feedback from you on whether our responses have adequately addressed your concerns.

Once again, we are grateful for your valuable contributions to our work.

Warm regards,

Authors of Submission 7027

---

### Meta-Review · Area_Chair_vcED · 2023-12-05

**Metareview:**

This paper presents a JsonTuning, a simple approach to improve the ability of large language models in following complex task instructions. Instead of using free-form input, JsonTunning uses structured JSON objects to represent task specifications. Via such structured task representations, JsonTunning decouples four types of information in the original free-form task specifications: (1) high-level task instructions (e.g., detect the language in text), (2) test examples (e.g., a text to detect its language), (3) candidate outputs (e.g., a list of candidate languages and their probabilities), (4) additional constraints and explanations on the outputs, such as their data types (e.g., language is of type string and its probability is a float).

The authors applied JsonTuning to five LLMs (LLaMA1/2-7B/13B and Falcon-7B) and compared with fine-tuning those LLMs using free-form text instructions, showing that JsonTuning yields LLMs that are better at **generalization**, able to understand different requirements in novel tasks. The models are also more **robust** against variations in the style of prompts and output formats, as well as being more **controllable** over the output structures, essential for solving tasks with complex output constraints such as relation extraction.

**Strengths:**

* The paper is well written and easy to read (pk2e, 5Thw, jpTc)

* The proposed approach that uses structured JSON objects as task specifications is straight-forward and intuitive, with nice empirical improvements over the text-tuning baselines  (qRB3, jpTc,5Thw). Reviewer jpTc also recognized that such a simple approach may have limited scientific impact, but this should be considered as reasons for rejection.

* The authors present comprehensive experiments and ablation studies over five LLMs (pk2e, qRB3, 5Thw). Still, some important ablations are missing in the initial submission, as detailed in weaknesses (comparison with text-tuning).

**Weaknesses:**

**Comparison with Text-tuning** as raised by reviewers pk2e and qRB3, the most important issue of this paper is the comparison with the text-tuning baseline, since JsonTuning leverages additional task-specific output constraints (“label space and control information” as mentioned in author responses), it’s essential to compare with a text-tuning baseline that uses such additional task-specific knowledge. The authors provided additional experiment results during the rebuttal phase, showing the gap between text-tuning (21.86 -> 23.50) and JsonTuning (25.37) is closing. However, this result is obtained only on one of the five LLMs (LLaMA-7b), and it remains unclear the difference between this more reasonable text-tuning baseline against JsonTuning for the other four models. Additionally, reviewer pk2e also raised a concern about the much worse results on TQA when JsonTuning does not use output constraints as compared to text-tuning (18.20 vs 10.40). The authors response, which hypothesizes that this is due to JsonTuning not being able to produce valid JSON structures without output constraints, seems concerning as it may suggest that JsonTuning could be more sensitive to output format errors for tasks with complex output structures (e.g., answers in TQA could be string or arrays or nested arrays) when such constraints are not available.

**Generalization to More Task Varieties** while JsonTuning is more effective for tasks with complex output structures, it might hurt performance on free-form, open-ended text generation tasks (qRB3),  as JSON still lags far behind natural language in terms of expressivity (5Thw). This issue was partially resolved in the rebuttal phase as the authors provided additional results on AlpacaEval featuring open-ended prompts, as requested by qRB3. However, the reviewers are still concerned about the extra overhead for real-world users to prompt LLMs with structured inputs instead of free-form text (qRB3), and the potential user-interaction challenges in real-world applications (5Thw).

**Extra Overhead to Encode JSON objects** compared to text-only tuning, JsonTuning requires additional input tokens to encode the boilerplate structure of JSON objects, which incurs additional computational overhead (jpTc, 5Thw). From the author's response,  JsonTuning uses approximately 25% more tokens for training and evaluation.

**Decouple improvements on generalization and controllability** (additional feedback from AC): most discussions in the papers based on averaged results across different evaluation datasets (MMLU, BBH, NER, RE, EE, TQA, NL2SQL), making it difficult to disentangle the gains from novel task generation vs controllability (two benefits of JsonTuning according to the Introduction). The authors may consider further grouping the evaluation datasets based on these two aspects. Still, they did a good job in robustness analysis in Section 3.3

**Justification For Why Not Higher Score:**

After the rebuttal, the issue regarding the comparison with text-based fine-tuning approaches (**weaknesses-1**) is still not fully addressed.  Therefore it's not clear if JsonTuning could outperform the test-base tuning baseline on all the LLMs evaluated in this paper. There is also an open issue regarding interpreting the worse results from JsonTuning without output structures, which may suggest that JsonTuning could be more sensitive to output format errors for tasks with complex output structures when such constraints are not available.

**Justification For Why Not Lower Score:**

N/A

---

### Decision · Program_Chairs · 2024-01-16

Reject